# Learning Human-Object Interaction as Groups

**Jiajun Hong**[*],    **Jianan Wei**[*],    **Wenguan Wang**[†]

Zhejiang University

https://github.com/JiajunHong1/GroupHOI

## Abstract

Human-Object Interaction Detection (HOI-DET) aims to localize human-object pairs and identify their interactive relationships. To aggregate contextual cues, existing methods typically propagate information across all detected entities via self-attention mechanisms, or establish message passing between humans and objects with bipartite graphs. However, they primarily focus on pairwise relationships, overlooking that interactions in real-world scenarios often emerge from collective behaviors (*i.e.*, multiple humans and objects engaging in joint activities). In light of this, we revisit relation modeling from a *group* view and propose GroupHOI, a framework that propagates contextual information in terms of *geometric proximity* and *semantic similarity*. To exploit the geometric proximity, humans and objects are grouped into distinct clusters using a learnable proximity estimator based on spatial features derived from bounding boxes. In each group, a soft correspondence is computed via self-attention to aggregate and dispatch contextual cues. To incorporate the semantic similarity, we enhance the vanilla transformer-based interaction decoder with local contextual cues from HO-pair features. Extensive experiments on HICO-DET and V-COCO benchmarks demonstrate the superiority of GroupHOI over the state-of-the-art methods. It also exhibits leading performance on the more challenging Nonverbal Interaction Detection (NVI-DET) task, which involves varied forms of higher-order interactions within groups.

## 1 Introduction

> *No man is an island, entire of itself.*
>
> *– John Donne, Meditation XVII Devotions*

Human-Object Interaction Detection (HOI-DET), as a critical pillar in visual relationship understanding, identifies entities (*i.e.*, humans and objects) as basic building blocks, and leverages relationships as the connective glue that weaves them into meaningful patterns. Early works [1, 2] typically recognize `HO-pairs`, which are cropped from natural images by manually obtained bounding boxes, as composites (*i.e.*, visual phrases) in isolation. Recent efforts [3, 4] extend HOI reasoning to real-world scenarios involving `multiple entities` with complex relational structures. Building upon object detection frameworks, HOI-DET methods evolve alongside advancements in detector architectures from Faster R-CNN [5] to DETR-like variants [6, 7]. As a semantic interpretation task, HOI-DET can also benefit from large visual-linguistic models (*e.g.*, CLIP [8] and BLIP [9]) pre-trained on extensive image-text corpora. Despite architectural innovations and multi-modal knowledge transfer, the core challenge remains unchanged: ❶*how to structure and reason about relationships among entities?*

Current methods [3, 10, 11] primarily reason over global context via the self-attention mechanism (Fig. 1(a)), while another strand of research constrains information exchange within homogeneous entities (*i.e.*, human-human, object-object) [12, 13] or heterogeneous neighborhoods (*i.e.*, human-object, object-human) [11] via bipartite graph (Fig. 1(b)). Despite achieving impressive performance,

---

\* The first two authors contribute equally to this work.

† Corresponding Author: Wenguan Wang.

39th Conference on Neural Information Processing Systems (NeurIPS 2025).

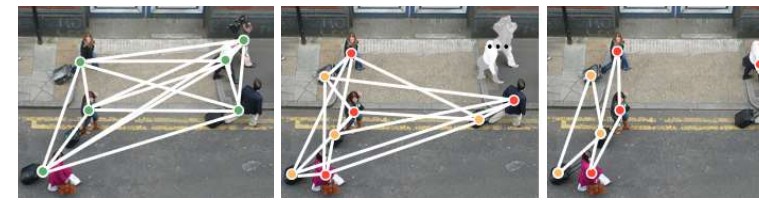

| (a) Complete Graph | (b) Bipartite Graph | (c) Geometric Graph | (d) Semantic Graph |

Figure 1: **Relation modeling paradigms**. While existing methods primarily model relations using complete (a) or bipartite (b) graphs, our approach introduces geometric (c) and semantic (d) graphs, enabling more structured and context-aware relational reasoning. ●, ●, and ● denote humans, objects, and interactions, respectively.

these methods remain confined to the modeling of predefined pairwise relations, leaving the inherent collective patterns [14] (*i.e.*, multiple entities engaging in joint activities) between entities unexplored.

Tackling this notable void brings us back to the classical clustering view for structuring and organizing visual entities by finding groups of similar ones, *i.e.*, **detecting HOIs as groups**. Tomasello's theory of shared intentionality [15] posits that with joint goals ("we-mode") and role coordination, participants tend to draw close to each other and perform collaborative actions. Considering a busy street with numerous pedestrians (Fig. 1), even without their explicit interpersonal relationships or social affiliations, the individuals can be naturally categorized into distinct groups based on their spatial proximity (Fig. 1(c)). Moreover, the walkers *pulling suitcases* typically exhibit similar visual characteristics—walking forward with swinging arms and suitcases trailing behind them— highlighting their shared behavior (Fig. 1(d)). Hence question ❶ becomes more fundamental from a group view: ❷ *how is a group formed?* and ❸ *how does a group function?*

Driven by question ❷, we construct social groups guided by the Gestalt grouping principles of Proximity and Similarity [16]: **First**, the ***geometric proximity principle***, *i.e.*, the tendency for individuals to form groups with those physically close to, is implemented by constructing *geometric groups* based on the central distance and Intersection over Union (IoU) of bounding boxes. **Second**, the ***semantic similarity principle***, *i.e.*, the tendency for individuals to affiliate with those sharing visual characteristics or behavioral patterns, is operationalized through building *semantic groups* of interaction proposals based on similarity of their feature embeddings. Notably, groups in social environments demonstrate complex and diverse compositions with overlapping memberships, which complicates quantification and modeling. To address this, we treat each proposal and its neighbors as a distinct group, establishing a one-to-one correspondence between groups and proposals.

To address question ❸, we propose GroupHOI, a HOI detector that enhances interaction reasoning by discovering inherent group patterns within visual scenes. Building on DETR [17], we model two distinct group structures to capture contextual dependencies: **First**, based on object detection outputs, we construct geometric groups for each entity and perform structure-aware reasoning via soft correspondence through self-attention, enabling fine-grained aggregation and targeted dissemination of local cues (§3.3). **Second**, we construct semantic groups to extract localized interaction priors, which are then integrated into the interaction decoder via a pooling operator to enable context-aware reasoning through a structured local-global processing scheme (§3.4). Visualizations in §4.5 provide transparency into how contextual dependencies are organized and leveraged.

To conclude, our contributions are: **i)** defining two visual attraction principles to construct groups; **ii)** proposing a one-stage framework that integrates these principles into HOI-DET; **iii)** endowing the HOI detector with enhanced interpretability via grouping mechanisms.

Our method is evaluated on two standard HOI-DET benchmarks: V-COCO [18] and HICO-DET [2], achieving impressive performance with **36.70** mAP on HICO-DET and **65.0** mAP on V-COCO. It surpasses the state-of-the-art method (*i.e.*, Pose-Aware [19]) by solid margins, *i.e.*, **+0.84** and **+3.9** mAP. Moreover, it achieves leading performance on the more challenging Nonverbal Interaction Detection (NVI-DET) [20] task, which permits diverse forms of higher-order group interactions. On the NVI benchmark [20], it reaches **73.19** AR on `val` and **75.21** AR on `test`.

## 2   Related Work

**Human-Object Interaction Detection.** Current HOI detectors can be categorized into two-stage and one-stage paradigms. Two-stage methods [1, 2, 21–28] typically employ off-the-shelf detectors (*e.g.*,

Faster R-CNN [5]) to locate humans and objects, then generate human-object pairs for interaction recognition. In contrast, one-stage methods [29–39], inspired by DETR [6], reformulate the task as a set prediction problem, and perform end-to-end triplets detection without explicit pairing. Recent studies [3, 4, 40–44] show strong gains by transferring knowledge from models pre-trained on large-scale image-text corpora (*e.g.*, CLIP [8], BLIP [9], and Stable Diffusion [45]) or large language models (*e.g.*, OPT [46]). Instead of focusing solely on architectural innovations or large-scale knowledge transfer, our approach highlights a complementary perspective on relation modeling, emphasizing how to organize and structure relations among entities.

**Relation Modeling for Detection Tasks.** Early object detectors [47–49] mainly built upon R-CNN family to localize and recognize each object in isolation, without contextual information exchange. To tackle this issue, [50] proposes a cascaded multi-stage framework with group recursive learning, while [51] incorporates global context across local regions. Recently, transformer-based methods such as DETR [6] reformulate detection as a set prediction problem, using self-attention to aggregate global context and enhance relational reasoning. HOI-DET methods have undergone a similar evolution. Early two-stage HOI-DET methods [23–25] confine information sharing to pre-defined human-object pairs and restrict interaction reasoning to isolated triplets, while DETR-based HOI-DET methods [30–32] leverage self-attention mechanisms to enable more comprehensive relational reasoning. Graph-based approaches [11, 13, 52–56] construct fully connected or bipartite graphs to facilitate message passing between entities. Building on this idea, several recent works [57, 58] further explore hypergraph representations, offering a more expressive way for high-order relations. However, most of them prioritize *how to reason about relations across entities* while neglecting *how to structure the relations among them*, thereby introducing noise from spurious correlations.

**Group Analysis for Vision Tasks.** Groups, as the fundamental units of human society, have become a pivotal analysis tool to understand complex human-centric scenes, with applications in crowd analysis, pedestrian trajectory prediction, and collective activity recognition. In crowd analysis, group-based methods [59] treat groups as atomic units of the scene, while individual-group methods [60] seek to leverage collective human information rather than processing them in isolation. Recently, PANDA [61] dataset enriched the task by integrating global context, high-resolution localized details, and temporal activity patterns, providing rich and hierarchical annotations. For pedestrian trajectory prediction, a number of existing approaches [57, 62–64] aim to improve multi-person tracking by incorporating group relationships. For instance, GroupNet [57] addresses the multi-agent trajectory prediction problem by leveraging multiscale hypergraphs to model both pairwise and group-level interactions more effectively. Collective activity recognition principally aims to capture the nuances of multi-human interactions in wide field-of-view scene, prompting many explorations of intra-group dynamics. For instance, ARG [65] tries to model both appearance and spatial dependencies between humans, and SAM [66] builds a dense relation graph and prunes it to a sparse one.

Inspired by these endeavors, we revisit HOI-DET from a group perspective, explore the visual principles underlying group formation, and propose a dedicated framework designed to operationalize these principles for HOI-DET, while introducing little extra computational overhead.

## 3 Methodology

### 3.1 Preliminary

Prevailing HOI-DET methods [3, 4, 19, 67, 68] employ a standardized DETR-based [69] pipeline for instance detection. Given an input image $\boldsymbol{I}$ with position embeddings $\boldsymbol{P}_e$, a visual encoder preceded by a CNN backbone is applied to extract features $\boldsymbol{V}_e$. Then, an instance decoder $\mathcal{D}_{ins}$ is employed to transform human $\boldsymbol{Q}_h$ and object $\boldsymbol{Q}_o$ queries into the outputs by retrieving the encoded features $\boldsymbol{V}_e$:

$$[\boldsymbol{Q}_h', \boldsymbol{Q}_o'] = \mathcal{D}_{ins}(\boldsymbol{V}_e, \boldsymbol{Q}_h, \boldsymbol{Q}_o), \tag{1}$$

where $\boldsymbol{Q}_h', \boldsymbol{Q}_o'$ are then processed into bounding boxes $\boldsymbol{B}_h, \boldsymbol{B}_o$ and object class labels $\boldsymbol{L}_o$. In the interaction branch, human $\boldsymbol{Q}_h'$ and object $\boldsymbol{Q}_o'$ outputs are exhaustively enumerated [33, 68] or directly associated [3, 67, 69] to form HO-pairs and interaction queries $\boldsymbol{Q}_{ins}$ for interaction classification.

### 3.2 Rethinking the Relation Modeling in HOI-DET

**Revisiting Transformer-based Relation Modeling.** Building upon DETR [69], most transformer-based HOI-DET models [3, 10, 11] perform global relational modeling among all the entity candidates

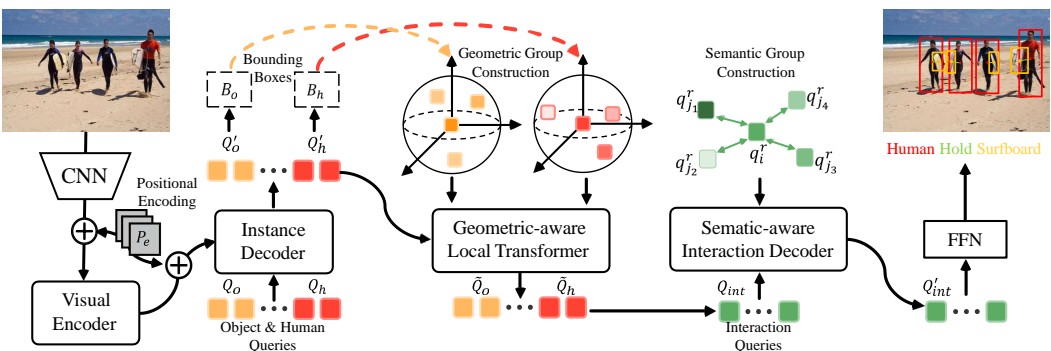

Figure 2: **Overview of GroupHOI**. GroupHOI comprises three key modules: **i)** a visual encoder extracts features, followed by an instance decoder to locate human-object pairs; **ii)** a geometric-aware local transformer aggregates and distributes contextual information within geometric groups; **iii)** a semantic-aware interaction decoder leverages local-global semantic dependencies for interaction prediction.

or their compositions via attention, which inherently establish a ***fully-connected graph*** for information exchange. Specifically, given an entity candidate $q_i^e \in \boldsymbol{Q}_e = \boldsymbol{Q}_h' \cup \boldsymbol{Q}_o'$, the relational modeling is achieved by the transformer decoder through self-attention operations with all entity queries:

$$\hat{\boldsymbol{q}}_i^e = \sum\nolimits_{\boldsymbol{q}_j^e \in \boldsymbol{Q}_e} \texttt{Softmax}(\boldsymbol{q}_i^e \cdot \boldsymbol{q}_j^e) \cdot \boldsymbol{q}_j^e, \tag{2}$$

where $\texttt{Softmax}$ propagates contextual cues by establishing soft correspondences from a global view, $\boldsymbol{q}_j^e \in \boldsymbol{Q}_e$ denotes all the detected entities.

**Revisiting Graph-based Relation Modeling.** Another line of research [11–13, 32, 54] constructs graphs based on the detected entities to enable relational modeling. Each node $\boldsymbol{q}_i^e$ is connected to a predefined set of neighboring nodes $\mathcal{N}_i$ and exchanges contextual information via message passing:

$$\hat{\boldsymbol{q}}_i^e = \boldsymbol{q}_i^e + \sigma(\sum\nolimits_{\boldsymbol{q}_j^e \in \mathcal{N}_i} \alpha_{ij} \cdot \texttt{MessagePassing}(\boldsymbol{q}_i^e, \boldsymbol{q}_j^e)), \tag{3}$$

where $\sigma$ is the activation function, $\alpha$ is an adjacency weight between nodes, and $\texttt{MessagePassing}$ denotes the message passing function. In terms of the set of neighboring nodes $\mathcal{N}(i)$, many studies [12, 13] define it as the union of all other entities in the scene, resulting in a ***fully-connected graph*** comprising both human and object nodes, while others [11, 13, 32, 54] restrict it to heterogeneous entities (*i.e.*, human-object, object-human), thereby formulating a ***bipartite graph***.

Though effective, these methods either treat all the entities as a single group or crudely partition them into two homogeneous groups. However, real-world scenarios demonstrate greater complexity in collective patterns: picture a cocktail party where a cluster of guests holding wine glasses or bottles is debating heatedly, while individuals seated on chairs around the venue are eating pizza. Given this divergence, we propose learning HOI as groups guided by the Gestalt grouping principles [16]: the *geometric proximity principle* and *semantic similarity principle*. The former is operationalized by *learning entities as geometric groups*, and the latter by *learning interactions as semantic groups*.

## 3.3 Learning Entities as Geometric Groups

According to Gestalt psychology [16], the proximity principle states that spatially adjacent elements are naturally perceived as coherent groups. This perceptual tendency has been applied as a strong inductive bias for structuring visual scenes [70]. Motivated by this principle, we extend it to HOI-DET by constructing geometric groups that organize detected entities based on their spatial configurations.

**Geometric Group Construction.** We construct groups for each entity embedding $\boldsymbol{q}_i^e$ by selecting its $K^g$ nearest neighbor entities based on the geometric distance of their corresponding bounding boxes $\boldsymbol{B}$ obtained by the instance decoder. However, simple geometric metrics often fall short in capturing the diverse sizes and spatial configurations of bounding boxes. To address this, we introduce a learnable estimator to measure geometric proximity among bounding boxes. First, we formulate the spatial feature $\boldsymbol{f}_{i,j}^p = [dis_{i,j}, IoU_{i,j}]$ by concatenating two geometric properties: **i)** the Euclidean distance $dis_{i,j}$ between the centroids of two bounding boxes $\boldsymbol{c}_i$ and $\boldsymbol{c}_j$ (*i.e.*, $dis_{i,j} = \sqrt{(\Delta \boldsymbol{c}_{i,j}^x)^2 + (\Delta \boldsymbol{c}_{i,j}^y)^2}$, where $\Delta \boldsymbol{c}_{i,j}^x = \boldsymbol{c}_i^x - \boldsymbol{c}_j^x$ and $\Delta \boldsymbol{c}_{i,j}^y = \boldsymbol{c}_i^y - \boldsymbol{c}_j^y$), and **ii)** the IoU between two

bounding boxes $\boldsymbol{B}_i$ and $\boldsymbol{B}_j$, i.e., $IoU_{i,j} = |\boldsymbol{B}_i \cap \boldsymbol{B}_j|/|\boldsymbol{B}_i \cup \boldsymbol{B}_j|$. Then, a simple linear layer is employed to compute the proximity score $s_{i,j}$ from $\boldsymbol{f}_{i,j}^p$ and yields a proximity score matrix. In this matrix, lower scores indicate closer spatial proximity. For each entity, we select the $K^g$ neighbors with the lowest scores to construct its geometric neighbor set $N_i^g$.

**Geometric Context Aggregation and Dispatch.** GroupHOI adopts a *geometric-aware local transformer* to capture the contextual cues from unordered entities by applying self-attention locally. First, to preserve the relative positional information within groups, we introduce trainable, parameterized position encodings $\boldsymbol{p}_{i,j}$ as follows:

$$\boldsymbol{p}_{i,j} = \delta(\boldsymbol{q}_i^e - \boldsymbol{q}_j^e), \qquad (4)$$

where $\delta$ is a two-layer MLP with ReLU nonlinearity. Then, we compute the dispatch matrix $\boldsymbol{t}_{i,j}$ for each entity:

$$\boldsymbol{t}_{i,j} = \texttt{Softmax}(\gamma(\phi_1(\boldsymbol{q}_i^e) - \phi_2(\boldsymbol{q}_j^e) + \boldsymbol{p}_{i,j})), \qquad (5)$$

where $\phi_1$ and $\phi_2$ are linear projections to perform point-wise feature transformation, and $\gamma$ is an MLP to produce the subtraction relation. Based on dispath matrix, GroupHOI adaptively aggregates the contextual cues within groups and utilizes them to update the entity embeddings:

$$\tilde{\boldsymbol{q}}_i^e = \theta(\textstyle\sum_{\boldsymbol{q}_j^e \in \mathcal{N}_i^g} \boldsymbol{g}_{i,j} \odot (\phi_3(\boldsymbol{q}_j^e) + \boldsymbol{p}_{i,j})) + \boldsymbol{q}_i^e, \qquad (6)$$

where $\phi_3$ is a linear projection, and $\theta$ denotes a feature alignment function. This process is applied among homogeneous entities, yielding $\tilde{\boldsymbol{Q}}_h$ and $\tilde{\boldsymbol{Q}}_o$ for human and object embeddings, respectively.

## 3.4 Learning Interactions as Semantic Groups

The concept of semantic grouping has long been recognized in cognitive science as a fundamental mechanism by which humans perceive and organize the world. According to Gestalt principles [16], humans naturally group elements based on contextual similarity to reduce cognitive load and enable efficient reasoning. Inspired by this insight, we incorporate semantic-aware grouping into interaction reasoning and enhance the vanilla transformer decoder with semantic-aware local cues.

**Semantic Group Construction.** We first initialize the interaction queries $\boldsymbol{Q}_{int}$ by computing the mean of $\tilde{\boldsymbol{Q}}_h$ and $\tilde{\boldsymbol{Q}}_o$. Given the complex and diverse group structures with overlapping memberships in social environments, we avoid using traditional partitioning techniques like $k$-means clustering to construct groups. Instead, we formulate distinct groups $\mathcal{N}_i^s$ for each interaction query $\boldsymbol{q}_i^r \in \boldsymbol{Q}_{int}$ by assessing pairwise cosine similarity $\boldsymbol{sim}_{ij} = (\boldsymbol{q}_i^r \cdot \boldsymbol{q}_j^r)/(||\boldsymbol{q}_i^r||||\boldsymbol{q}_j^r||)$ between it and other queries $\boldsymbol{q}_j^r$ and selecting its top-$K^s$ most similar counterparts.

**Semantic Context Aggregation.** Given each query $\boldsymbol{q}_i^r$ with its semantic group $\mathcal{N}_i^s$, we utilize max pooling to aggregate semantic message from its group members $\boldsymbol{q}_j^r$:

$$\boldsymbol{m}_i = \texttt{max}(\phi_4(\boldsymbol{q}_i^r, \boldsymbol{q}_j^r - \boldsymbol{q}_i^r)), \quad \boldsymbol{q}_j^r \in \mathcal{N}_i^s, \qquad (7)$$

where $\phi_4$ represents a sequence of linear layers, batch normalization, and ReLU layers.

**Local-Global Integration.** As described in §3.2, the transformer-based interaction decoder models global relations through self-attention but lacks explicit local relation modeling. To address this limitation, we construct a *semantic-aware interaction decoder* by integrating the proposed local semantic context aggregation mechanism into the naive decoder, which enhances its capacity for interaction reasoning. Specifically, prior to each decoder layer, the local semantic context is aggregated and incorporated into the original query via residual connection:

$$\hat{\boldsymbol{q}}_i^r = \boldsymbol{q}_i^r + \phi_5(\boldsymbol{m}_i), \qquad (8)$$

where $\phi_5$ shares the same identity structure as $\phi_4$. $\hat{\boldsymbol{q}}_i^r$ is subsequently passed through the self-attention and cross-attention modules in the interaction decoder layer.

## 3.5 Implementation Details

**Network Architecture.** To ensure fair comparison with previous HOI-DET methods [3, 4, 67, 71], we adopt ResNet-50 [72] as the visual backbone. Our transformer-based architecture consists of a 6-layer encoder, a 3-layer instance decoder, and a 3-layer interaction decoder. Following [3, 4], we initialize 64 learnable queries for human and object branches, and set the feature dimensions to

256 for human/object representations and 768 for interaction representations. We perform group construction independently at each layer of both the instance and interaction decoders, where the geometric and semantic group sizes are set to 4 and 2. We evaluate three variants of GroupHOI using different pre-trained VLMs: **i)** CLIP (ViT-B/16) [8], **ii)** CLIP (ViT-L/14), and **iii)** BLIP2 (ViT-L) [9]. In each variant, we follow [3, 4] to initialize the classification heads using VLM text embeddings and integrate visual features from the VLM visual encoder into the interaction decoder.

**Training Objectives.** Following [3, 4], our model is optimized by the following loss:

$$\mathcal{L}_{\text{HOI}} = \lambda_b \mathcal{L}_b + \lambda_u \mathcal{L}_u + \lambda_c^o \mathcal{L}_c^o + \lambda_c^a \mathcal{L}_c^a, \tag{9}$$

where $\mathcal{L}_b$ denotes the box regression loss, $\mathcal{L}_u$ indicates the intersection-over-union loss, $\mathcal{L}_c^o$ and $\mathcal{L}_c^a$ represent the cross-entropy loss for object and interactions classification, respectively. The coefficient factors $\{\lambda_b, \lambda_u, \lambda_c^o, \lambda_c^a\}$ are empirically set as $\{2.5, 1, 1, 1\}$.

**Reproducibility.** GroupHOI is implemented in PyTorch. The model is trained with a batchsize of 8 for 90 epochs on 2 GeForce RTX 4090 GPUs.

# 4 Experiment

## 4.1 Experiments on HOI-DET

**Datasets.** We conduct experiments on V-COCO [18] and HICO-DET [2]:

- **V-COCO** is a specialized subset of MS-COCO [73], which comprises 10,346 images (5,400 for training and 4,946 for testing). The dataset annotates 263 unique human-object interactions, covering 29 action categories and 80 object categories.
- **HICO-DET** comprises a total of 47,776 images, with 38,118 designated for training and 9,658 for testing. It includes 80 object categories, consistent with those in V-COCO dataset, 117 action categories and 600 distinct HOI classes.

**Evaluation Metrics.** We employ mAP as our evaluation metric. For V-COCO [18], mAP is reported under two different scenarios: scenario 1 for all 29 action categories, and scenario 2 which excludes 4 body motion categories. For HICO-DET [2], we evaluate GroupHOI in complete 600 HOI categories (Full), the 138 rare categories with fewer than 10 training instances (Rare), and remaining 462 categories (Non-Rare). For each object, mAP is computed in the whole dataset (Default) and subset containing the object (Known Object).

**Training.** Our backbone is initialized with DETR [6] pre-trained on MS-COCO [73]. The model is trained using AdamW optimizer [74] with a batchsize of 8 for 90 epochs. The initial learning is set to $5e^{-5}$, which reduces by a factor of 10 every 30 epochs.

**Testing.** For fairness, GroupHOI operates without data augmentation. We retain the top-$K$ HOI candidates ($K = 100$) followed by Non-Maximum Suppression for redundancy removal.

**Quantitative Results on HICO-DET.** As shown in Table 1, our model with a VIT-B/16 CLIP [8] variant already outperforms all the competitors under the same setting by a substantial margin. In particular, GroupHOI surpasses the previous state-of-the-art Pose-Aware [19] by **0.84/2.38/0.40** mAP on the Full, Rare, and Non-Rare settings. Notably, GroupHOI exhibits a comparable number of parameters and FLOPs to HOICLIP [4] (§4.3), but delivers substantial improvements of **2.11/3.74/1.52** mAP on HICO-DET [2]. Moreover, leveraging more powerful VLMs, *i.e.*, ViT-L/14 CLIP [8] and ViT-L BLIP2 [9], GroupHOI boosts its performance and sustains state-of-the-art results. Both with ViT-L/14 CLIP, GroupHOI suppresses CMMP[68] by a large margin with **1.32** mAP under Full, while it narrowly trails CMMP in Rare setting. Compared to UniHOI [40] which retrieves knowledge from large language models, GroupHOI with ViT/L BLIP2 still leads by **0.47/ 0.71/0.39** mAP.

**Quantitative Results on V-COCO.** For V-COCO, GroupHOI exhibits competitive performance but remains slightly inferior to STIP [33], VIL [75], and CQL [76]. However, on HOI-DET, our method surpasses them by a solid margin. We attribute this to GroupHOI's improved capability in handling more complex interactions, as evidenced by: HOI-DET contains 600 HOI interactions (*vs*.293 in V-COCO), an average of 4 interactions (*vs*.3 in V-COCO) and 6 entities per sample (*vs*.4 in V-COCO).

Table 1: **Quantitative results** on HICO-DET [2] `test` and V-COCO [18] `test`. DF and KO denote the Default and Known Object evaluation settings of HICO-DET. CLIP/B16, CLIP/B32, CLIP/L14 represent VIT-B/16, VIT-B/32 and VIT-L/14 settings for CLIP respectively. See §4.1 for details.

| Method | Config | HICO-DET (DF) | | | HICO-DET (KO) | | | V-COCO | |
|---|---|---|---|---|---|---|---|---|---|
| | | Full | Rare | Non-Rare | Full | Rare | Non-Rare | $AP_{role}^{S1}$ | $AP_{role}^{S2}$ |
| QPIC[67][CVPR21] | R50 | 29.07 | 21.85 | 31.23 | 31.68 | 24.14 | 33.93 | 58.8 | 61.0 |
| QPIC[67][CVPR21] | R101 | 29.90 | 23.92 | 31.69 | 32.38 | 26.06 | 34.27 | 58.3 | 60.7 |
| HOTR[69][CVPR21] | R50 | 23.46 | 16.21 | 25.60 | - | - | - | 55.2 | 64.4 |
| CDN(S)[17][NeurIPS21] | R50 | 31.44 | 27.39 | 32.64 | 34.09 | 29.63 | 35.42 | 61.2 | 63.8 |
| CDN(B)[17][NeurIPS21] | R50 | 31.78 | 27.55 | 33.05 | 34.53 | 29.73 | 35.96 | 62.3 | 64.4 |
| CDN(L)[17][NeurIPS21] | R101 | 32.07 | 27.19 | 33.53 | 34.79 | 29.48 | 36.38 | 63.9 | 65.9 |
| MSTR[77][CVPR22] | R50 | 31.17 | 25.31 | 32.92 | 34.02 | 28.83 | 35.57 | 62.0 | 65.2 |
| STIP[33][CVPR22] | R50 | 32.22 | 28.15 | 33.43 | 35.29 | 31.43 | 36.45 | **65.1** | **69.7** |
| PViC[71][ICCV23] | R50 | 34.69 | 32.14 | 35.45 | 38.14 | 35.38 | 38.97 | 62.8 | 67.8 |
| Pose-Aware[19][CVPR24] | R50 | **35.86** | **32.48** | **36.86** | **39.48** | **36.10** | **40.49** | 61.1 | 66.6 |
| DOQ[78][CVPR22] | R50+CLIP/B16 | 33.28 | 29.19 | 34.50 | - | - | - | 63.5 | - |
| GEN-VLKT[3][CVPR22] | R50+CLIP/B16 | 33.75 | 29.25 | 35.10 | 37.80 | 34.76 | 38.71 | 62.4 | 64.4 |
| ADA-CM[79][ICCV23] | R50+CLIP/B16 | 33.80 | 31.72 | 34.42 | - | - | - | 56.1 | 61.5 |
| HOICLIP[4][CVPR23] | R50+CLIP/B32 | 34.59 | 31.12 | 35.74 | 37.61 | 34.47 | 38.54 | 63.5 | 64.8 |
| VIL[75][ACMMM23] | R50+CLIP/B16 | 34.21 | 30.58 | 35.30 | 37.67 | 34.88 | 38.50 | 65.3 | 67.7 |
| CQL[76][CVPR23] | R50+CLIP/B16 | 35.36 | 32.97 | 36.07 | - | - | - | **66.4** | **69.2** |
| LOGICHOI[38][NeurIPS23] | R50+CLIP/B16 | 35.47 | 32.03 | 36.22 | 38.21 | 35.29 | 39.03 | 64.4 | 65.6 |
| CMMP[68][ECCV24] | R50+CLIP/B16 | 33.24 | 32.26 | 33.53 | - | - | - | - | 61.2 |
| HOIGen[68][ACMMM24] | R50+CLIP/B16 | 34.84 | 34.52 | 34.94 | - | - | - | - | - |
| CEFA[80][ACMMM24] | R50+CLIP/B16 | 35.00 | 32.30 | 35.81 | 38.23 | 35.62 | 39.02 | 63.5 | - |
| GroupHOI (ours) | R50+CLIP/B16 | **36.70** | **34.86** | **37.26** | **39.42** | **37.78** | **39.91** | 65.0 | 66.0 |
| ADA-CM[79][ICCV23] | R50+CLIP/L14 | 38.40 | 37.52 | 38.66 | - | - | - | 58.6 | 64.0 |
| UniVRD[81][ICCV23] | R50+CLIP/L14 | 37.41 | 28.90 | 39.95 | - | - | - | - | - |
| CMMP[68][ECCV24] | R50+CLIP/L14 | 38.14 | **37.75** | 38.25 | - | - | - | - | 64.0 |
| GroupHOI (ours) | R50+CLIP/L14 | **39.46** | 37.10 | **40.16** | **41.58** | **39.42** | **42.40** | **66.4** | **67.3** |
| UniHOI[40][NeurIPS23] | R50+BLIP2 | 40.06 | 39.91 | 40.11 | 42.20 | 42.60 | 42.08 | 65.6 | **68.3** |
| GroupHOI (ours) | R50+BLIP2 | **40.53** | **40.62** | **40.50** | **42.70** | **42.92** | **42.64** | **66.4** | 67.8 |

## 4.2 Experiments on NVI-DET

**Task Formulation.** Given an input image, it predicts a set of ⟨*individual*, *group*, *interaction*⟩ triplets, aiming to localize each individual and identify the social group it belongs to, while determining the category of its nonverbal interaction. Unlike HOI-DET [18] that focuses on recognizing actions between human-object pairs, NVI-DET models interactions in a more flexible and generalized manner. Specifically, it accounts for interactions involving arbitrary numbers of humans, which makes the task significantly more challenging.

**Dataset.** NVI [20] densely labeling social groups in pictures, along with 22 atomic-level nonverbal behaviors (16 individual- and 6 group-wise) under five broad interaction types. It contains 13,711 images in total and splits them into 9,634, 1,418 and 2,659 for train, val and test.

**Evaluation Metrics.** Consistent with [20], we utilize mean Recall@K (mR@K) for evaluation. Specifically, we report mR@25, mR@50, and mR@100 under different IoU thresholds for matching predictions with ground truth, along with their average (AR) to provide a comprehensive evaluation.

**Quantitative Results.** Following [20], we compare GroupHOI with modified versions of three state-of-the-art HOI-DET methods (*i.e.*, $m$-QPIC [67], $m$-CDN [17], and $m$-GEN-VLKT [3]) and NVI-DEHR [20] on NVI [20]. As shown in Table 2, GroupHOI surpasses all other methods, reaching **73.19** and **75.21** AR on NVI `val` and `test`. Furthermore, we analyze performance across individual-wise and group-wise interactions. Table 3 shows our model consistently outperforms others by significant margins in both sets. These results verify our model's effectiveness in identifying social group structures and capturing the collective behaviors within them.

## 4.3 Diagnostic Experiments

As shown in Table 4, we conduct a set of ablation studies on HICO-DET [2] for deeper analysis.

Table 2: **NVI-DET results** on NVI [20] `val` and `test`. See §4.2 for details.

| Method | val | | | | test | | | |
|---|---|---|---|---|---|---|---|---|
| | mR@25 | mR@50 | mR@100 | AR | mR@25 | mR@50 | mR@100 | AR |
| $m$-QPIC[67][CVPR21] | **56.89** | 69.52 | 78.36 | 68.26 | 59.44 | 71.46 | 80.07 | 70.32 |
| $m$-CDN[17][NeurIPS21] | 55.57 | 71.06 | 78.81 | 68.48 | 59.01 | 72.94 | 82.61 | 71.52 |
| $m$-GEN-VLKT[3][CVPR22] | 50.59 | 70.87 | 80.08 | 67.18 | 56.68 | 74.32 | 84.18 | 71.72 |
| NVI-DEHR[20][ECCV24] | 54.85 | 73.42 | 85.33 | 71.20 | 59.46 | 76.01 | **88.52** | 74.67 |
| GroupHOI (ours) | 55.67 | **76.73** | **87.16** | **73.19** | **62.67** | 76.57 | 86.42 | **75.21** |

Table 3: **Individual-** and **group-wise** interactions results on NVI [20] `val`. See §4.2 for details.

| Method | individual | | | | group | | | |
|---|---|---|---|---|---|---|---|---|
| | mR@25 | mR@50 | mR@100 | AR | mR@25 | mR@50 | mR@100 | AR |
| $m$-QPIC[67][CVPR21] | **52.23** | 66.09 | 75.98 | 64.77 | 69.18 | 78.62 | 84.85 | 77.55 |
| $m$-CDN[17][NeurIPS21] | 50.67 | 68.23 | 76.74 | 65.21 | 68.66 | 78.60 | 84.34 | 77.20 |
| $m$-GEN-VLKT[3][CVPR22] | 44.98 | 68.51 | 78.30 | 63.93 | 67.84 | 79.47 | 87.12 | 78.14 |
| NVI-DEHR[20][ECCV24] | 49.37 | 70.04 | 83.82 | 67.74 | 69.47 | 82.45 | 89.35 | 80.42 |
| GroupHOI (ours) | 49.60 | **74.03** | **85.63** | **69.75** | **71.87** | **83.92** | **91.24** | **82.34** |

**Analysis of Key Modules.** We analyze the effects of geometric and semantic groups in our framework. As shown in Table 4 (a), the results indicate that: **First**, both geometric and semantic relation learning contribute to HOI prediction. Incorporating geometric and semantic groups individually yields mAP gains of **0.67/0.91/0.81** and **0.21/0.45/0.31** under Full/Rare/Non-Rare settings respectively. **Second**, the combination of them yields optimal performance, delivering **0.98/2.66/0.70** mAP improvements over the baseline, which highlights a more comprehensive relation modeling in HOI prediction.

Table 4: **Ablation study** of GroupHOI on HICO-DET [2] `test`, where GEO and SEM represent geometric group and semantic group, *Homo.* and *Hetero.* mean homogeneous group and heterogeneous group (§4.3).

(a) Analysis of key modules

| GEO | SEM | Full | Rare | Non-Rare |
|---|---|---|---|---|
| - | - | 35.72 | 32.20 | 36.56 |
| ✓ | - | 36.39 | 33.11 | 37.37 |
| - | ✓ | 35.93 | 32.65 | 36.87 |
| ✓ | ✓ | **36.70** | **34.86** | **37.26** |

(b) Analysis of geometric group size

| Group Size | Full | Rare | Non-Rare |
|---|---|---|---|
| $K^g = 2$ | 36.23 | 32.97 | 37.13 |
| $K^g = 3$ | 36.34 | 33.87 | 37.32 |
| $K^g = 4$ | **36.70** | **34.86** | **37.26** |
| $K^g = 5$ | 36.45 | 33.57 | 36.95 |

(c) Analysis of semantic group size

| Group Size | Full | Rare | Non-Rare |
|---|---|---|---|
| $K^s = 1$ | 36.07 | 32.09 | 37.26 |
| $K^s = 2$ | **36.70** | **34.86** | **37.26** |
| $K^s = 3$ | 36.18 | 32.67 | 37.21 |
| $K^s = 4$ | 35.92 | 32.06 | 37.07 |

(d) Analysis of fusion strategy

| Model | Full | Rare | Non-Rare |
|---|---|---|---|
| Baseline | 35.72 | 32.20 | 36.56 |
| + *Homo.* | 36.23 | 32.40 | 37.20 |
| + *Hetero.* | **36.70** | **34.86** | **37.26** |

(e) Analysis of geometric group layer

| Group Layer | Full | Rare | Non-Rare |
|---|---|---|---|
| $L^g = 1$ | **36.70** | **34.86** | **37.26** |
| $L^g = 2$ | 36.47 | 31.96 | 37.81 |
| $L^g = 3$ | 35.64 | 31.20 | 37.00 |
| $L^g = 4$ | 36.17 | 32.72 | 37.20 |

(f) Analysis of semantic group layer

| Group Layer | Full | Rare | Non-Rare |
|---|---|---|---|
| $L^s = 1$ | 36.13 | 31.58 | 37.49 |
| $L^s = 2$ | 36.53 | 32.20 | 37.82 |
| $L^s = 3$ | **36.70** | **34.86** | **37.26** |
| $L^s = 4$ | 36.13 | 31.53 | 37.51 |

**Analysis of Group Size.** We evaluate the impact of the geometric group size $K^g$ and semantic group size $K^s$ in Table 4 (b) and (c). Our design performs independent group construction at each decoder layer, allowing entities and interactions to exchange information with different neighbors across layers. As shown in the table, the performance peaks at $K^g = 4$ and drops as $K^g$ increases, suggesting that oversized geometric groups introduce noise from less relevant neighbors. In contrast, the model performs best at $K^s = 2$, and both larger and smaller $K^s$ degrade performance, implying only highly relevant semantic cues are beneficial for interaction prediction. Notably, since groups are constructed independently per layer, the geometric and semantic group sizes across the three-layer decoder range from 4-10 and 2-4, respectively, under the optimal setup. This aligns with real-world scenarios: over 94% of samples in V-COCO [18] and 91% in HICO-DET [2] involve no more than 10 entities, and even large groups are dominated by a few key participants (only around 4) [82].

**Homo. *vs*. Hetero. Geometric Group.** We make a comparison between homogeneous and heterogeneous paradigms for geometric groups in Table 4 (d). The heterogeneous paradigm models humans and objects as distinct node types with exclusive intra-class (human-human/object-object) message passing. In contrast, the homogeneous paradigm treats all entities uniformly, allowing unrestricted cross-entity communication. The heterogeneous design achieves higher performance, surpassing the homogeneous counterpart by **0.47/2.46/0.06** mAP. This disparity suggests that integrating humans and objects into a group may dilute relevant context and hinder the specialized information exchange.

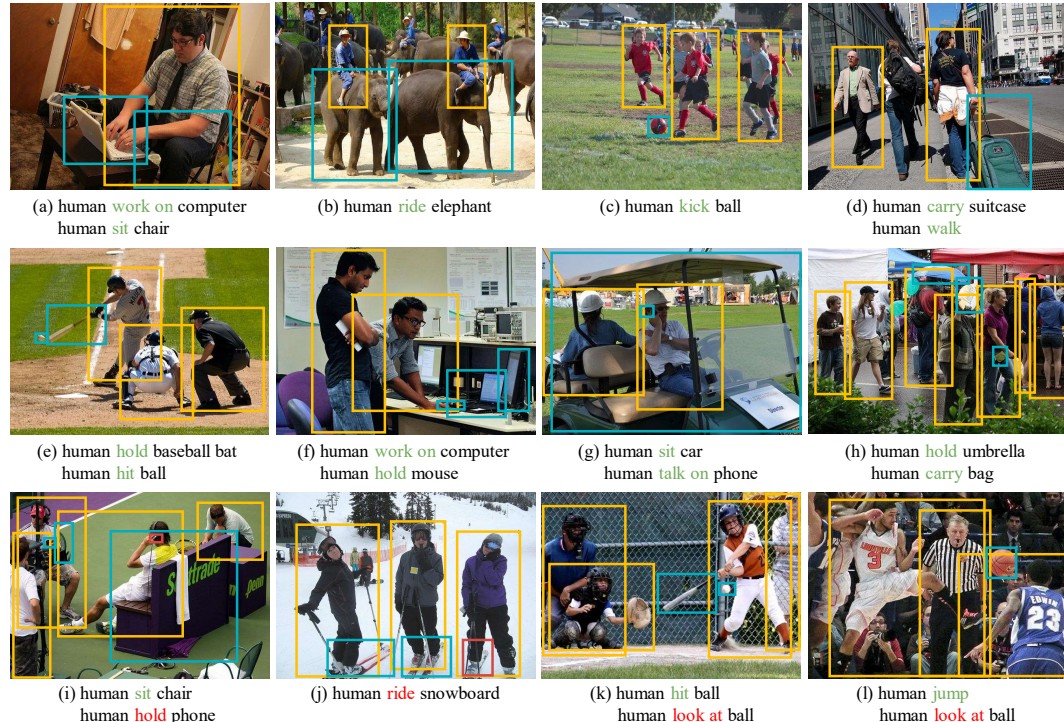

(a) human work on computer
human sit chair

(b) human ride elephant

(c) human kick ball

(d) human carry suitcase
human walk

(e) human hold baseball bat
human hit ball

(f) human work on computer
human hold mouse

(g) human sit car
human talk on phone

(h) human hold umbrella
human carry bag

(i) human sit chair
human hold phone

(j) human ride snowboard

(k) human hit ball
human look at ball

(l) human jump
human look at ball

Figure 3: **Visualization of GroupHOI results** on V-COCO [18] `test`. The first two rows represent samples where all interactions are successfully detected, while the third row corresponds to samples with missed interactions. Detected interactions are marked in Green, while missed interactions and objects are in Red.

**Analysis of Group Layer.** Table 4 (e) and (f) present the evaluation of geometric group layer $L^g$ and semantic group layer $L^s$. We observe that increasing $L^g$ does not lead to consistent improvement. While the performance improves as $L^s$ increases from 1 to 3, after which the gains plateau, indicating that excessive local context aggregation brings diminishing returns.

**Comparison of Model Efficiency.** Table 5 compares the model scalability in terms of the number of Parameters (Params), Floating Point Operations (FLOPs), and Frames Per Second (FPS). Although GroupHOI maintains a comparable parameter count and experiences only a marginal reduction in inference speed compared to HOICLIP [4], it achieves significantly better performance, out-

Table 5: Comparison of **model efficiency**.

| Method | Params↓ | FLOPs↓ | FPS↑ |
|---|---|---|---|
| PPDM[31][CVPR20] | 194.9M | 121.63G | 15.28 |
| GEN-VLKT[3][CVPR22] | 41.9M | 60.04G | 26.18 |
| HOICLIP[4][CVPR23] | 66.1M | 104.68G | 19.57 |
| ViPLO[83][CVPR23] | 118.2M | 41.35G | 14.89 |
| GroupHOI (ours) | 79.2M | 83.67G | 16.42 |

performing HOICLIP by **2.11** mAP on HICO-DET [2] (see Table 1). Compared to ViPLO [83], GroupHOI achieves superior performance (**+1.75** mAP) while requiring fewer parameters and less inference time, underscoring its efficiency. A detailed comparison is available in Table S2 in Appendix.

## 4.4 Qualitative Results

Fig. 3 illustrates the qualitative results of GroupHOI on V-COCO [18] `test`. As seen, it can robustly localize and predict interactions under challenging conditions. For instance, our method generalizes well to interactions with little training data like *riding elephant* in Fig. 3(b), consistent with its superior rare-set performance (Table 1). As shown in Fig. 3(c-h), GroupHOI also handles complex scenes with multiple humans and objects, benefiting from effective intra-group information propagation. The third row presents failure cases, mainly due to: **i)** severe occlusion causing missed detections, *e.g.*, the failure in detecting *human hold phone* (Fig. 3(i)) and *human ride skateboard* (Fig. 3(j)). **ii)** Ambiguous interactions arising from insufficient spatiotemporal information, such as difficulty in recognizing gaze direction (Fig. 3(k)). **iii)** Lack of domain-specific knowledge, exemplified by the misclassification of *looking at ball* as *kicking ball* in (Fig. 3(l)), which is caused by lacking knowledge of basketball, *i.e.*, players typically do not use their feet to kick the ball during games. These

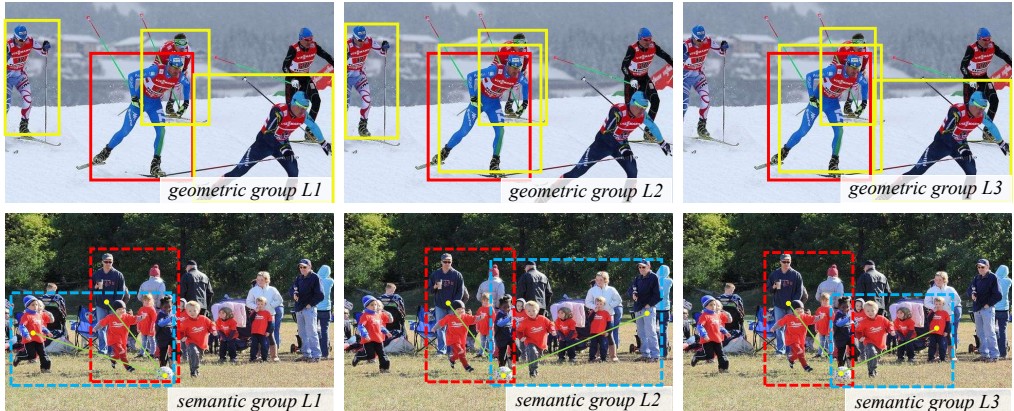

Figure 4: **Visualization of groups** on V-COCO [18] `test`. L1, L2 and L3 represent the first, second and last layer of instance or interaction decoder. The center of the geometric and semantic groups is mark in Red (§4.5).

limitations could potentially be addressed by developing spatiotemporal reasoning modules based on video-level datasets [84] or integrating knowledge via pre-trained large language models [40].

## 4.5 Visualization of Groups

Fig. 4 shows examples of geometric and semantic groups across decoder layers on V-COCO [18] `test`. In the first row, two main geometric group patterns emerge: **i)** multiple proposals targeting to the same human/object, enabling feature completion; **ii)** adjacent distinct proposals that interact closely to show mutual influence. Moreover, groups may also include unannotated entities, revealing GroupHOI's ability to exploit implicit contextual relations. When grouping by semantic similarity (the second row of Fig. 4), the grouping patterns can also be observed, *i.e.*, distinct individuals performing the same interaction. In summary, our method demonstrates the capability to automatically uncover latent patterns in complex scenarios, thereby enhancing the model's holistic scene comprehension.

## 5 Discussion

**Future Direction.** Current HOI benchmarks involve limited entities within small fields of view, leading to global relational modeling in the mainstream HOI-DET [18] methods. However, when applied to larger scenarios (*e.g.*, gigapixel-level crowd images [61]), this paradigm leads to high computational cost and information redundancy. A practical solution is to confine relational modeling to local regions, making the principles proposed in this paper highly applicable to real-world settings.

**Limitation.** A limitation of our method is its focus on interaction reasoning without integrating the proposed mechanisms into the object detection branch. We think this direction intriguing, yet it lies beyond the scope of the present study. Moreover, like other HOI-DET [18] models, GroupHOI is trained with limited label diversity, posing challenges for generalization to in-the-wild scenarios.

## 6 Conclusion

We present GoupHOI, a framework builds upon the idea of reorganizing the unordered in the visual scenes by exploring their inherent grouping patterns. By revisiting HOI-DET [18] task from a group perspective, we formulate two visual attraction principles, *i.e.*, *geometric proximity* and *semantic similarity*, to explain group dynamics. These principles are instantiated via two paradigms: **i)** learning entities as geometric groups, and **ii)** learning interactions as semantic groups. By introducing marginal computational overhead, GroupHOI advances state-of-the-art methods by solid margins and sets new SOTAs for HOI-DET and NVI-DET [20] task, which verifies its superiority.

**Acknowledgement.** This work was supported by National Natural Science Foundation of China (No. 62372405), Fundamental Research Funds for the Central Universities (226-2025-00057), Zhejiang Provincial Natural Science Foundation of China (No. LD25F020001), and CIE-Tencent Robotics X Rhino-Bird Focused Research Program.

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

## Summary of the Appendix

For a better understanding of the main paper, we provide additional details in this supplementary material, which is organized as follows:

- §A provides more implementation details of GroupHOI.
- §B offers more qualitative results.
- §C provides more experimental results.
- §D presents the pseudo code of GroupHOI.
- §E discusses the societal impact of this work.

## A    More Implementation Details

We follow HOICLIP [4] in converting HOI triplets and object labels into textual descriptions to generate CLIP [8] text embeddings. Specifically, each HOI triplet <human, verb, object> is converted into a sentence like "A photo of a person [verb-ing] a/an [object]". For "no-interaction" instances, we use "A photo of a person and a/an [object]", and object labels are converted into "A photo of a/an [object]". These textual descriptions are then processed by the pre-trained CLIP text encoder to obtain corresponding embeddings, which initialize the weights of the interaction classifier $\mathcal{C}^a$ and object classifier $\mathcal{C}^o$. During training, these classifiers are fine-tuned with a small learning rate to adapt to the specific dataset. We employ the pre-trained CLIP visual encoder to extract visual features $V_{clip}$. These features, along with our encoded features $V_e$, are independently processed by separate interaction decoders. The resulting outputs are then fused to facilitate the final reasoning. We also introduce two key modifications based on HOICLIP [4]. **First**, to address the dimensional mismatch between positional embeddings (256) and the interaction decoder (768), we expand the positional embeddings by stacking them three times. **Second**, we replace the Focal Loss with Asymmetric Loss [85] to better handle the long-tail distribution of HOI categories.

## B    More Qualitative Results

We further provide qualitative examples of our approach in Fig. S1. These results highlight GroupHOI's robust performance in HOI detection across various scenes. Notably, our model effectively captures complex interaction patterns in scenarios involving group activities such as team sports (*e.g.*, soccer or tennis). This capability enhances interaction prediction with mutual communication. We also present several failure examples of our model, primarily due to missed object detection, as seen in Fig. S1(c). Additionally, our model encounters challenges when dealing with highly ambiguous relations. For instance, in Fig. S1(s), GroupHOI fails to detect the *look at ball* between the girl at the center and the ball, which is disrupted by her surrounding teammates.

## C    More Experiments

**Ablative Experiments.** We evaluate four strategies for measuring geometric proximity between entities, using intersection-over-union (IoU), center distance (CD), and global image features (IF). As shown in Table S1, combining IoU and CD consistently yields the best performance across all splits, indicating that both spatial cues complement each other effectively. In contrast, adding global image features slightly degrades performance, suggesting that they are not essential for proximity estimation.

Table S1: Analysis of geometric proximity measurement on HICO-DET [2] test(§C).

| Measurement | Full | Rare | Non-Rare |
|---|---|---|---|
| *IOU only* | 36.02 | 32.19 | 37.16 |
| *CD only* | 36.14 | 33.60 | 36.90 |
| *IOU + CD* | **36.70** | **34.86** | **37.26** |
| *IOU + CD +IF* | 36.21 | 33.45 | 36.97 |

**Efficiency Comparison.** Table S2 presents a comprehensive comparison of the model scalability (*i.e.*, number of Parameters (Params), Floating Point Operations (FLOPs) and Frames Per Second (FPS)) for various HOI detection methods. As shown, GroupHOI achieves significant performance improvements over previous models while maintaining a comparable number of parameters. We also

Table S2: Comparison of efficiency and performance on HICO-DET [2] `test` and V-COCO [18] `test`.

| Method | Backbone | Params↓ | FLOPs↓ | FPS↑ | Default Full | Default Rare | Default Non-Rare | $AP_{role}^{S1}$ | $AP_{role}^{S2}$ |
|---|---|---|---|---|---|---|---|---|---|
| iCAN[22][BMVC18] | R50 | 39.8 | - | - | 14.84 | 10.45 | 16.15 | 45.3 | - |
| DRG[54][ECCV20] | R50-FPN | 46.1 | - | - | 19.26 | 17.74 | 19.71 | 51.0 | - |
| PPDM[31][CVPR20] | HG104 | 194.9 | - | - | 21.73 | 13.78 | 24.10 | - | - |
| SCG[32][ICCV21] | R50-FPN | 53.9 | - | - | 31.33 | 24.72 | 33.31 | 54.2 | 60.9 |
| HOTR[69][CVPR21] | R50 | 51.2 | - | - | 25.10 | 17.34 | 27.42 | 55.2 | 64.4 |
| HOITrans[37][CVPR21] | R50 | 41.4 | - | - | 23.46 | 16.91 | 25.41 | 52.9 | - |
| AS-Net[86][CVPR21] | R50 | 52.5 | - | - | 28.87 | 24.25 | 33.14 | 53.9 | - |
| QPIC[67][CVPR21] | R50 | 41.9 | - | - | 29.07 | 21.85 | 31.23 | 58.8 | 61.0 |
| CDN-S[17][NeurIPS21] | R50 | 42.1 | - | - | 31.78 | 27.55 | 33.05 | 62.3 | 64.4 |
| STIP[33][CVPR22] | R50 | 50.4 | - | - | 32.22 | 28.15 | 33.43 | **65.1** | **69.7** |
| GEN-VLKT[3][CVPR22] | R50 | 41.9 | 60.04 | 26.18 | 33.75 | 29.25 | 35.10 | 62.4 | 64.4 |
| HOICLIP[4][CVPR23] | R50 | 66.1 | 104.68 | 19.57 | 34.59 | 31.12 | 35.74 | 63.5 | 64.8 |
| CLIP4HOI[87][NeurIPS23] | R50 | 71.2 | - | - | 35.33 | 33.95 | 35.74 | - | 66.3 |
| ViPLO[83][CVPR23] | ViT-B/32 | 118.2 | - | - | 34.95 | 33.83 | 35.28 | 60.9 | 66.6 |
| GroupHOI (ours) | R50 | 79.2 | 83.67 | 16.42 | **36.70** | **34.86** | **37.26** | 65.0 | 66.0 |

compare FLOPs and FPS with GEN-VLKT [3] and HOICLIP [4]. Despite a marginal reduction in inference speed relative to HOICLIP, GroupHOI has lower FLOPs and yields solid mAP improvements of **2.11/3.74/1.52** in HICO-DET [2], highlighting the effectiveness of our proposed framework.

# D  Pseudo Code

The pseudo code for semantic and geometric group are given in Algorithm 1 and Algorithm 2.

**Algorithm 1:** Pseudo-code for Geometric Group in a PyTorch-like style.

```
"""
hs: output human/object embeddings from the instance decoder.
pos_embed: position embeddings for human/object queries.
coords: bounding boxes of humans/objects.
K_g: geometric group size.
"""
def Geometric_Group(hs, pos_embed, coords, K_g):
    # Formulate the spatial feature
    F_p = Cat([Square_distance(coords[:, None], coords[None, :]), IoU(coords[:, None], coords[None, :])])
    # Compute the proximity score
    S = Linear(F_p)
    # Select the topk neighbors
    knn_idx = TopK(S, K_g)

    # Compute the position encodings
    pos_enc = MLP(pos_embed - Gather(pos_embed, knn_idx))
    # Formulate query, key, and value
    q, k, v = Linear(hs), Linear(Gather(hs, knn_idx)), Linear(Gather(hs, knn_idx))
    # Compute dispatch matrix
    G = Softmax(q - k + pos_enc)
    # Aggregate geometric context
    C_g = Linear(G * (v + pos_enc))
    out = hs + C_g

    return out
```

**Algorithm 2:** Pseudo-code for Semantic Group in a PyTorch-like style.

```
"""
hs: interaction embeddings from the interaction decoder.
K_s: semantic group size.
"""
def Semantic_Group(hs, pos_embed, K_s):
    # Compute the similarity score
    S = CosineSimilarity(hs[:, None], hs[:, None])
    # Select the topk neighbors
    knn_idx = TopK(S, K_s)

    # Aggregate semantic context
    C_s = MLP(Max(MLP(hs[:, None], hs[:, None]-Gather(hs, knn_idx)), dim=-1))
    out = hs + C_s

    return out
```

# E Boarder Impact

This work advances the recognition of human-object interactions in complex scenarios, particularly in scenes where small groups of people naturally form, which is a common occurrence in real-world settings. This capability holds significant promise for applications in collaborative robotics, autonomous systems, healthcare monitoring, among others. However, there are also potential downsides. Our method risks propagating irrelevant contextual information among entities that merely happen to be co-located but share no collective pattern, leading to "hallucinated" interaction predictions. Moreover, group-level clustering may inadvertently propagate systemic biases, particularly in scenarios requiring differential treatment of individuals within clusters (*e.g.*, unfair reward and punishment allocation in crowd behavior analysis). Hence, it is essential to rigorously consider legal regulations and integrate certain fairness constraints to avoid potential negative societal impacts.

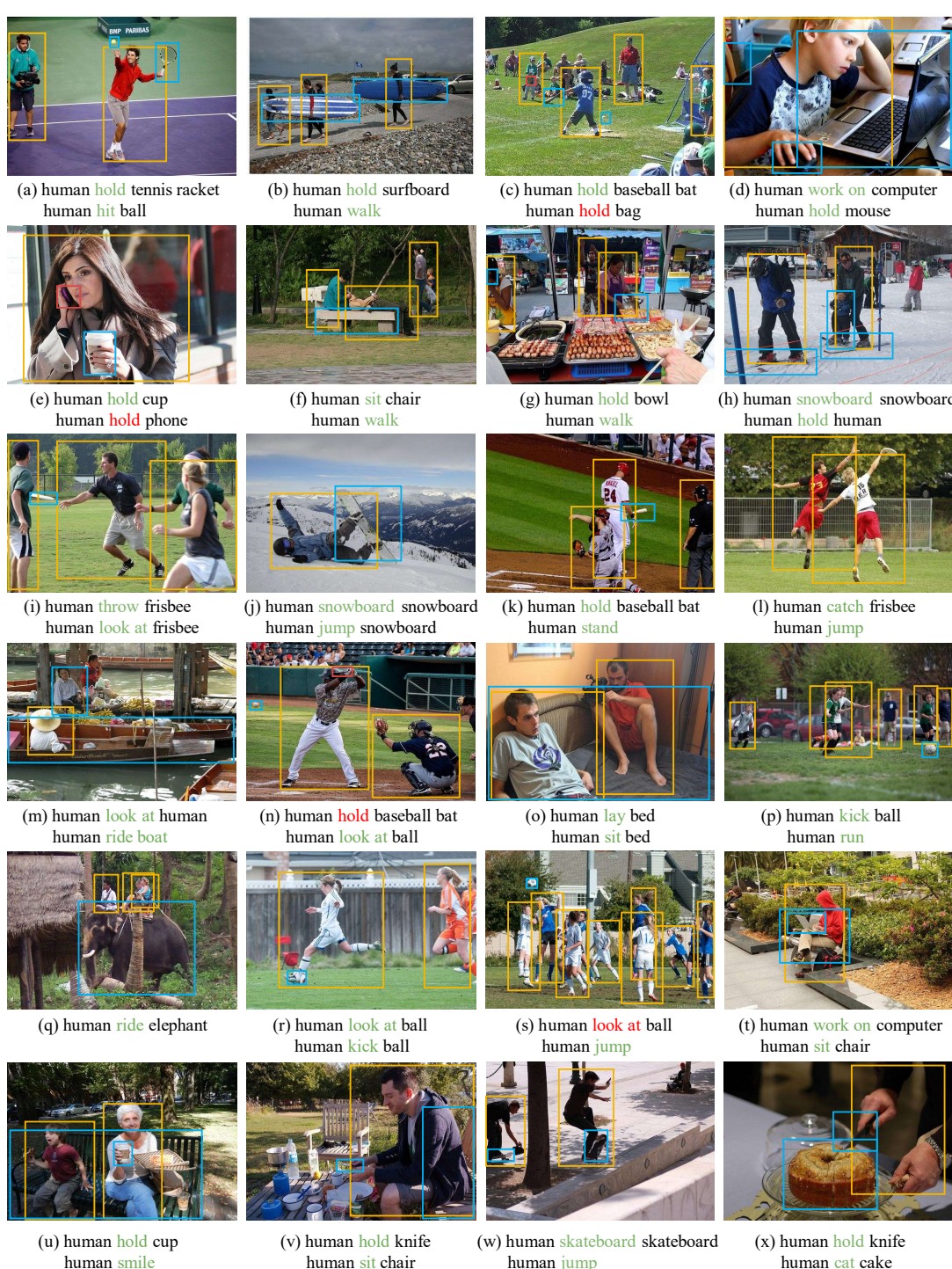

Figure S1: Visualization of GroupHOI results on V-COCO [18] test. Detected interactions are marked in Green, while missed interactions and objects are in Red.

