# OpenReview forum: "Learning Human-Object Interaction as Groups"
_NeurIPS.cc/2025/Conference — NeurIPS 2025 poster_

### Official Review · Reviewer_8R1p · 2025-06-09

**Clarity:** 3
**Significance:** 2
**Originality:** 3
**Rating:** 4
**Confidence:** 4

**Summary:**

This paper introduces GroupHOI for Human-Object Interaction (HOI) detection that models interactions not just as pairs but as groups, inspired by social grouping principles. The method forms geometric groups based on spatial proximity and semantic groups based on feature similarity, leveraging both for improved contextual reasoning. GroupHOI integrates these principles into a DETR-based architecture, enhancing interaction prediction by propagating information within learned groups. Experiments on the HICO-DET and V-COCO benchmarks show GroupHOI achieves state-of-the-art results.

**Questions:**

Why does the GroupHOI set the size of the geometric and semantic groups to 4 and 2, while the example in Figure 4 uses a size of 3? Does this mean that group sizes of 4 and 2 may not be the most appropriate for some images?

**Ethical Concerns:**

["NO or VERY MINOR ethics concerns only"]

**Final Justification:**

The rebuttal addressed my concerns. However, since the proposed method GroupHOI is built upon a modified and stronger version of HOICLIP, the reported relative improvements appear somewhat overstated. As a result, I still consider this paper to be at the borderline for acceptance.

**Limitations:**

See weaknesses.

**Quality:**

2

**Strengths And Weaknesses:**

Strengths:
1. The joint geometric and semantic perspective to model complex relationships between entities and propagate contextual information across them is both innovative and interesting.
2. The paper defines two visual principles for constructing geometric and semantic groups, and effectively integrates these principles into HOI detection.
3. Overall writing of this paper is good and easy to understand and follow.

Weaknesses:
1. The authors did not provide evidence to demonstrate that the proposed *geometric and semantic perspective groups* can address the current challenges faced by existing HOI methods. It remains unclear whether these groups can cover all possible HOI scenarios, and whether the method is effective in handling cases that do not fit the situations described in the paper.
2. The size of geometric and semantic groups are fixed. Since the content of each image varies, this fixed number of sizes conflicts with the assumptions made in the paper.
3. Although the model achieves SOTA performance, the source of the improvement is unclear. As shown in Table 2(a), the baseline without GEO and SEM already outperforms CMMP and LOGICHOI. While it is undeniable that GroupHOI improves upon the baseline, Table 2(a) suggests that the performance gain of GroupHOI may stem more from other factors rather than the GEO and SEM modules.

---

> ### Author Rebuttal · Authors · 2025-07-31
>
> We thank reviewer 8R1p for the valuable time and constructive feedback. We provide point-to-point response below.
>
> ***
>
> **Q1: Insufficient Evidence for Addressing HOI Challenges**
>
> **A1**: Respectively disagree. We would like to clarify that our work have clearly delineated the limitations of existing HOI methods and demonstrated how our proposed approach specifically addresses these challenges:
>
> - **Challenge**. We explicitly define the challenge addressed in this paper as **how to model the complex relationships between entities and propagate contextual information across them?** (see L32-33).
>
> - **Limitation**. ​​For this challenge, current methods remain confined to the modeling of predefined pairwise relations, leaving the inherent collective patterns (i.e., multiple entities engaging in joint activities) between entities unexplored (see L34-40).
>
> - **Motivation**. To tackle this limitation，we revist HOI detection from a group perspective, i.e., structuring and organizing visual entities by finding groups of similar ones, formulating the challenge as two foundamental issues: (1) **how is a group formed?** (2) **how does a group function?** (see L41-50).
>
> - **Technical Solution**. For group formulation, we propose two grouping strategies based on geometric proximity and semantic similarity principles (see L51-60). For group functioning, we introduce a new architecture for HOI detection involving these two grouping strategies (see L61-72).
>
> - **Experiment**. ​​The superior performance of GroupHOI on V-COCO and HICO-DET demonstrates its ability to address the core challenge targeted by this study (see L76-78).
>
> Overall, we argue that our geometric proximity and semantic similarity groups can effectively address the challenge raised by this study.
>
> **Q2:  Whether these groups can cover all possible HOI scenarios, and whether the method is effective in handling cases that do not fit the situations described in the paper.**
>
> **A2**: It seems a little unfair that requires our method to cover all possible HOI scenarios, as no existing dataset provides comprehensive coverage for such validation. However, in response to your concern, we introduce additional validation on NVI dataset [ref1] which incorporates complex interactions involving arbitrary numbers of humans ranging from 2 to 12. Following [ref1], we report mR@25, mR@50, and mR@100 under different IoU thresholds. The experimental results on NVI val are as follows:
>
> | Method              | mR@25     | mR@50     | mR@100    | AR        |
> | :------------------ | :-------- | :-------- | :-------- | :-------- |
> | m-QPIC              | 56.89     | 69.52     | 78.36     | 68.26     |
> | m-CDN               | 55.57     | 71.06     | 78.81     | 68.48     |
> | m-GEN-VLKT          | 50.59     | 70.87     | 80.08     | 67.18     |
> | NVI-DEHR [ECCV24]   | 54.85     | 73.42     | 85.33     | 71.20     |
> | **GroupHOI (Ours)** | **55.67** | **76.73** | **87.16** | **73.19** |
>
> As seen, GroupHOI surpasses all these methods, reaching new sota for NVI-DET task. This result demonstrates that our model is applicable to more complex interaction scenarios.
>
> [ref1] Nonverbal Interaction Detection. ECCV 2024.
>
>
> **Q3: Why is the group size fixed?**
>
> **A3**: Sorry for this confusion. Both the geometry-aware local transformer and the semantic-aware interaction decoder adopt a three-layer Transformer decoder architecture. In each decoder layer, a distinct geometric or semantic grouping is established to model contextual relationships. This strategy enables the total number of neighbors for each proposal (i.e., human, object, and interaction proposal) ranging from $K_g$ to $K_g\times3$ (**4 to 12**) for the geometric group and from $K_s$ to $K_s\times3$ (**2 to 6**) for the semantic group. Notably, over 96% of samples in V-COCO and over 94% in HICO-DET contain no more than 12 entities, so geometric group of 4 to 12 can cover the majority of scenarios. Meanwhile, the size of semantic group aligns with [ref2]'s finding, which underscores that the majority of the interactive dialogue involving 10 or more participants is produced by only the top 4-5 contributors. The above discussion will be incorporated into Sec. 3.5 and 4.3.
>
> [ref2] Group discussion as interactive dialogue or as serial monologue: The influence of group size. Psychological science, 2000.
>
> **Q4: Source of baseline's performance gain**
>
> **A4**: We apologize for the omission of implementation details. Our baseline is built upon HOICLIP [ref3], with its performance primarily attributed to the following modifications:
>
> - **Training strategy**. We maintain the same training duration (​​90 epochs​​) as HOICLIP. However, our initial learning rate is set to $5\times10^{-5}$​​, decaying by a factor of ​​10 after every 30 epochs​​. In contrast, HOICLIP uses an initial learning rate of $1\times10^{-4}$​, reducing it by a factor of ​​10 only after 60 epochs​​.
> - **HOI/object classifier**. Unlike HOICLIP, we do not freeze the parameters of the HOI/object classifiers initialized from CLIP, allowing for further optimization during training.
> - **Interaction decoder**. Due to the dimension mismatch, HOICLIP omit positional embeddings (256) in the interaction decoder (768). Instead, we expand the positional embeddings to 768 by stacking them three times for the interaction decoder.
> - **HOI loss**. To better address the long-tail distribution issue, we replace the original Focal loss with Asymmetric Loss [ref4] for HOI loss computation.
>
> The aforementioned information will be included in the Appendix Sec. A.
>
> [ref3] Hoiclip: Efficient knowledge transfer for hoi detection with vision-language models. CVPR 2023.
>
> [ref4] Asymmetric loss for multi-label classification. ICCV 2021.
>
> **Q5: Examples in Fig. 4.**
>
> **A5**: Apologies for the confusion. As stated in **Q3**, we construct a new geometric/semantic group at each interaction decoder layer, yielding geometric group sizes of **4-12** and semantic group sizes of **2-6**. Thus, to facilitate presentation, we select three entities from the group for visualization. We apologize for the ambiguity in our previous statement. To clarify, we will update the visualization in Fig. 4 to illustrate the evolving group composition at each layer in the revised manuscript.

---

> > ### Comment · Reviewer_8R1p · 2025-08-05
> >
> > Thank you for your rebuttal, which has partially addressed my concerns. I will raise my score. However, since the proposed method GroupHOI is built upon a modified and stronger version of HOICLIP, the reported relative improvements appear somewhat overstated. As a result, I still consider this paper to be at the borderline for acceptance.

---

> > > ### Author Response · Authors · 2025-08-05
> > > **Thanks for the response**
> > >
> > > Thank you for your thoughtful review and for taking the time to consider our rebuttal. We are delighted to hear that the additional information addressed your concerns, prompting a rise in the rating. Your feedback is invaluable in improving the quality of our paper and we are committed to incorporating your suggestions for a new version that reflects the changes.

---

### Official Review · Reviewer_PSyr · 2025-06-29

**Clarity:** 3
**Significance:** 3
**Originality:** 3
**Rating:** 4
**Confidence:** 4

**Summary:**

This paper proposes a method for Human-Object Interaction detection that explicitly models interactive relationships within a group. To exploit these relationships, the authors propose constructing geometric and semantic groups based on geometric proximity and semantic similarity. A modified transformer architecture is also introduced to perform feature aggregation and distribution within each group. Experimental results demonstrate the effectiveness of the proposed method.

**Questions:**

- How does the learnable estimator ensure the accuracy of group classification?
- How does the semantic group account for category differences?

**Ethical Concerns:**

["NO or VERY MINOR ethics concerns only"]

**Final Justification:**

All of my concerns have been addressed. I hope the additional experiments and discussions can be included in the final version of the paper.

**Limitations:**

yes

**Paper Formatting Concerns:**

N.A

**Quality:**

3

**Strengths And Weaknesses:**

Strengths
- Considering the relationships within groups to promote HOI is an effective and intuitive idea.
- The authors exploit the principle of proximity to form groups, which adheres to real-world human behavior and is conceptually sound.

Weaknesses
- The author proposes a learnable estimator to measure the proximity among bounding boxes. Does this module require supervision? Without supervision, how can the MLPs distinguish whether two entities belong to the same group? How much improvement can be achieved by using MLPs over conventional distance-based calculations for estimating group? Additionally, the MLPs only take distance and IoU as inputs, which may not be sufficient to determine whether two entities belong to the same group—for example, two people who are far apart in depth might still have overlapping bounding boxes. Therefore, should the input also incorporate image features?
- The group sizes within the same image can vary significantly, but this paper does not account for the variation in group size. Simply setting the geometric and semantic group numbers to 4 and 2 does not reflect the actual variability in real-world scenarios.
- There are previous works that have explored leveraging intra-group information to understand interactions. They also form groups based on proximity and similarity, and their structures are not strictly limited to fully-connected or bipartite graphs. Although these works do not specifically address the HOI problem, they share many similarities with this work in terms of human group construction, and thus, I believe they should be discussed as well.

     [A] "Reconstructing groups of people with hypergraph relational reasoning." Proceedings of the IEEE/CVF international conference on computer vision. 2023.

     [B] "Groupnet: Multiscale hypergraph neural networks for trajectory prediction with relational reasoning." Proceedings of the IEEE/CVF Conference on Computer Vision and Pattern Recognition. 2022.
- In the Semantic Group, there may be significant differences between human and object features. When computing feature similarity, is it possible that only humans are grouped together while objects are excluded from the group?

---

> ### Author Rebuttal · Authors · 2025-07-31
>
> We thank reviewer PSyr for the valuable time and constructive feedback. We provide point-to-point response below.
>
> ***
>
> **Q1: Does the learnable proximity estimator require supervision? If not, how does it distinguish groups?**
>
> **A1**: Apologies for the confusion. The linear layer, used to compute the proximity score by fusing the centroid distance and IoU between two bounding boxes, is not explicitly supervised but is trained in an end-to-end manner based on the final HOI loss. The proximity score measures the spatial closeness between two detection boxes, where a smaller value indicates closer proximity. We construct the neighbor set by selecting the $K_g$ detection boxes with the smallest proximity scores. To clarify, the manuscript will be revised in Sec. 3.3 as follows:
>
> > A simple linear layer is then employed to compute the proximity score $s_{i,j}$ based on the central distance $dis_{i,j}$ and the Intersection-over-Union $IoU_{i,j}$ between bounding boxes, without requiring additional supervision. This results in a proximity score matrix, where lower scores indicate closer spatial proximity. For each entity, we select the $K_g$ neighbors  with the lowest scores to construct its geometric neighbor set $N^g_i$.
>
> **Q2: What is the performance gain of MLPs over distance-based methods?**
>
> **A2**: In response to your suggestion, we have added the ablation experiments, as:
>
> | Method              | Full (DF) | Rare (DF) | Non-Rare (DF) |
> | ------------------- | --------- | --------- | ------------- |
> | GroupHOI (IOU only) | 36.02     | 32.19     | 37.16         |
> | GroupHOI (CD only)  | 36.14     | 33.60     | 36.90         |
> | **GroupHOI (Ours)** | **36.70** | **34.86** | **37.26**     |
>
> where (IoU only) and (CD only) refer to using the IoU and center distance for group construction, respectively. Results demonstrate the linear fusion layer employed in GroupHOI provides a significant performance advantage.
>
> **Q3: Are distance and IoU sufficient for grouping, or should image features be included?**
>
> **A3**: Thank you for the insightful suggestion. We agree that incorporating depth maps [ref1] could be a promising direction to address the limitation you identified. However, it would introduce additional supervision and computational overhead. Moreover, it could lead to an unfair comparison with existing methods that rely solely on 2D visual cues.
>
> Following your suggestion, we also conducted additional experiments utilizing image features:
>
> | Method                    | Full (DF) | Rare (DF) | Non-Rare (DF) |
> | :------------------------ | :-------- | :-------- | :------------ |
> | GroupHOI (image features) | 36.21     | 33.45     | 36.97         |
> | **GroupHOI** **(Ours)**   | **36.70** | **34.86** | **37.26**     |
>
> For GroupHOI (image features), we incorporate image features into the computation of the proximity score by concatenating them with two geometric metrics. However, its results showed no significant improvements. We hypothesize that the image features in the HOI detector are also insensitive to depth, as the labels do not provide such supervisory signals.
>
> [ref1] Improving visual relation detection using depth maps. ICPR, 2021.
>
> **Q4:  Fixed group size.**
>
> **A4**: We acknowledge the confusion and apologize. Both our geometry-aware local transformer and semantic-aware interaction decoder employ a three-layer Transformer decoder architecture. For each decoder layer, **we construct a new geometric/semantic group**. Through this strategy, each proposal (i.e., human, object, and interaction proposal) can aggregate contextual information from $K_g$ to $K_g\times3$ (**4 to 12**) for the geometric group and from $K_s$ to $K_s\times3$ (**2 to 6**) for the semantic group. Empirical statistics show that over 96% of samples in V-COCO and over 94%  in HICO-DET contain no more than 12 entities, ensuring that our geometric grouping strategy sufficiently covers the majority of scenarios. The semantic group design is motivated by [ref2], which observes that even in large-scale interactive settings, meaningful exchanges are often dominated by a small subset of participants (4–5 key contributors), aligning with our design. The above discussion will be incorporated into Sec. 3.5 and 4.3.
>
> [ref2] Group discussion as interactive dialogue or as serial monologue: The influence of group size. Psychological science, 2000.
>
>
> **Q5: Discussion about related prior group-based methods.**
>
> **A5**: Thank you for your valuable suggestion. We agree that these works share similarities with our work and will add a discussion of them in Sec. 2 as follows:
>
> > Recent studies in related domains have also explored group-wise relational modeling based on proximity and similarity. For example, [ref3] addresses 3D human mesh recovery in crowded scenes using a multiscale hypergraph to capture both individual and group-level relations. Similarly, [ref4] focuses on multi-agent trajectory prediction by learning multiscale hypergraphs to represent pairwise and group-wise interactions. These works  share the idea of modeling intra-group relations with spatial or semantic cues. However, they simply combine proximity and similarity information for relation reasoning (e.g., [ref3] introduces human spatial information through box coordinates). In contrast, our method explicitly constructs and learns relationships from both geometric and semantic perspectives through two distinct graphs: a geometric graph based on the spatial proximity of entities, and a semantic graph built on the semantic similarity of interactions. This design enables more fine-grained relation structuring tailored to HOI scenarios.
>
> [ref3] Reconstructing groups of people with hypergraph relational reasoning. ICCV, 2023.
>
> [ref4] Groupnet: Multiscale hypergraph neural networks for trajectory prediction with relational reasoning. CVPR, 2022.
>
>
> **Q6: Semantic Group for human and object features.**
>
> **A6**: We apologize for the confusion. We construct geometric groups exclusively for human and object proposals (see L162-172), and semantic groups exclusively for interaction proposals (see L192-197) initialized by computing the mean of human and object proposals. Additionally, in Sec. 4.3, we empirically observed that distinguishing human and object proposals is necessary for constructing geometric groups (see Lines 278–281, Table 5c).

---

> > ### Comment · Reviewer_PSyr · 2025-08-05
> >
> > Thank you for the response. The experiments on image features are quite interesting, and all of my concerns have been addressed. I hope the additional experiments and discussions can be included in the final version of the paper.

---

> > > ### Author Response · Authors · 2025-08-05
> > > **Thanks for the response**
> > >
> > > Thank you for your thoughtful review and for taking the time to consider our rebuttal. We are delighted to hear that the additional information addressed your concerns, prompting a rise in the rating. Your feedback is invaluable in improving the quality of our paper and we are committed to incorporating your suggestions for a new version that reflects the changes.

---

### Official Review · Reviewer_H8t4 · 2025-06-29

**Clarity:** 4
**Significance:** 3
**Originality:** 3
**Rating:** 5
**Confidence:** 4

**Summary:**

This paper introduces GroupHOI to enhance human-object interaction (HOI) understanding by modeling relationships from a group perspective rather than pairwise ones. GroupHOI is built upon two grouping principles:
- Proximity principle: This principle suggests that individuals tend to form groups with those they are physically close to. In GroupHOI, a learnable proximity estimator is introduced to measure this. Contextual cues within these groups are aggregated and update the entity embeddings.
- Semantic similarity principle: This principle suggests that interactions sharing similar visual or behavioral patterns tend to form semantic groups. GroupHOI implements this by forming clusters of interaction proposals (i.e., human-object pairs) based on cosine similarity of their feature embeddings. These groups are then used to provide semantic context to the interaction reasoning process, and local features within bounding boxes are also incorporated.

The framework adopts a DETR-based cascade architecture, in line with other recent state-of-the-art approaches.

GroupHOI demonstrates performance improvements over existing methods on standard HOI benchmarks, including HICO-DET and V-COCO.

**Questions:**

- From the examples shown in Figure 4, it appears that some nearest neighbors in the geometric grouping may actually refer to the same entity due to overlapping detection boxes (e.g., before non-maximum suppression is applied). In such cases, how often do self-matches occur when selecting the top K_g nearest entities? Do you feel necessary to address the self-matches?

- Moreover, since object instances can vary significantly in category, size, and spatial layout, why use fixed neighbors? Would it be more flexible to allow the learnable proximity estimator to determine the appropriate number of neighbors dynamically, rather than using a fixed top-k scheme?

- For the geometric group construction, how is the proximity score S_{ij} trained? What kind of supervision signal is used to guide the learning of this estimator? Is there a loss that explicitly supervises the group formation process, or is it trained in an end-to-end manner solely based on the final HOI loss in Eq 9?

**Ethical Concerns:**

["NO or VERY MINOR ethics concerns only"]

**Final Justification:**

The authors have provided answers for my questions to clarify my misunderstandings.

**Limitations:**

yes

**Paper Formatting Concerns:**

no concerns.

**Quality:**

3

**Strengths And Weaknesses:**

**Strengths**

- The paper is clearly written and well-organized. The authors clearly articulate the motivation for introducing grouping in HOI tasks and provide clear explanations of their corresponding design choices. The overall presentation reflects strong experience in writing research papers.

- The central idea of modeling relationships from a group perspective is technically sound. The proposed solution is coherent and conceptually aligned with the proposals.

- The experimental results, including ablations, are comprehensive and provide convincing support for the proposed method. Overall, the paper slightly exceeds my acceptance threshold.

**Weaknesses**

- Overall, I think the selected datasets such as HICO-DET and V-COCO may not be ideal for showcasing the benefits of group-based relationship modeling. These datasets have been used extensively for years, and most recent works report only marginal improvements. It's unclear how much of the remaining error is attributable to the absence of group-level reasoning, as opposed to dataset-specific limitations such as annotation bias, limited scale, or inherent noise. Additionally, the majority of interactions in these datasets likely involve single human–single object pair, which may limit the opportunity for group modeling techniques to demonstrate clear advantages.

- A more compelling direction could be for the authors to curate or design a new benchmark that emphasizes scenarios requiring group-level reasoning, for example, interactions involving multiple humans and objects with interdependent roles. Alternatively, the authors could consider filtering or reorganizing the existing datasets to introduce a new split of complex multi-entity interactions, going beyond the typical rare/non-rare splits.

---

> ### Author Rebuttal · Authors · 2025-07-31
>
> We thank reviewer H8t4 for the valuable time and constructive feedback. We provide point-to-point response below.
>
> ----
>
> **Q1: HOI-DET and V-COCO may not be ideal for group-based relationship modeling.**
>
> **A1**: Thanks for your thoughtful suggestion! Although HICO-DET and V-COCO contain abundant single human-object interaction instances (e.g., a photo of a person sitting on a chair), we find that there is still a significant proportion of samples that contain multiple human-object pairs via statistical analysis. Specifically, HICO-DET demonstrates an average of four interactions per image, while V-COCO contains three interactions per image. However, to address your concern, **we have supplemented our experiments with results on NVI datasets [ref1], which encompasses complex interactions involving multiple humans** (see Q2). Additionally, we strongly agree with your perspective and intend to develop a new visual relationship benchmark for group-based relationship modeling in our future work.
>
> [ref1] Nonverbal Interaction Detection. ECCV, 2024.
>
>
> **Q2: Design or adopt a benchmark for group-level reasoning.**
>
> **A2**: We appreciate the insight! We conduct additional experiments on NVI datasets [ref2], which encompasses complex interactions involving multiple humans. These experiments incorporate the Nonverbal Interaction Detection (NVI-DET) task, designed to localize individual entities and their corresponding social groups while identifying the category of its nonverbal interaction. The experimental results on NVI val are as follows:
>
> | Method              | mR@25     | mR@50     | mR@100    | AR        |
> | :------------------ | :-------- | :-------- | :-------- | :-------- |
> | m-QPIC              | 56.89     | 69.52     | 78.36     | 68.26     |
> | m-CDN               | 55.57     | 71.06     | 78.81     | 68.48     |
> | m-GEN-VLKT          | 50.59     | 70.87     | 80.08     | 67.18     |
> | NVI-DEHR [ECCV24]   | 54.85     | 73.42     | 85.33     | 71.20     |
> | **GroupHOI (Ours)** | **55.67** | **76.73** | **87.16** | **73.19** |
>
> These results demonstrate that our method also generalizes well to more complex interaction scenarios. We will  add Sec. 4.4 for the analysis of results on NVI dataset [ref2]  in the revised manuscript.
>
> [ref2] Nonverbal Interaction Detection. ECCV, 2024.
>
>
> **Q3: How often do self-matches occur, and should they be addressed?**
>
> **A3**: As you noted, self-match can occur in both geometric and semantic groups (see Fig. 4). However, it is difficult to quantify the frequency of self-matches in the groups formulated by GroupHOI: with a fixed number of proposals (see L217) in our method, self-matches increase as detectable entities in an image decrease, and decrease as detectable entities increase. This scenario is not considered necessary to address in our method for the following reasons:
>
> - Detection boxes typically exhibit deviations from ground truth ones; therefore, the overlapping detection boxes corresponding to the same entity can mutually complement their information by self-matches.
> - For inputs with limited detectable entities (e.g., images only containing single human-object interaction), the groups formulated by GroupHOI are dominated by self-matches, which enables our method to maintain certain performance (albeit weaker) under such constraints.
>
>
> **Q4: Why adopt fixed neighbors？Is it possible to train an estimator for neighbor count assessment?**
>
> **A4**: My apologies for the misunderstanding. Both our geometry-aware local transformer and semantic-aware interaction decoder employ a three-layer Transformer decoder architecture. For each decoder layer, **we construct a new geometric/semantic group**. This strategy enables the total number of neighbors for each proposal (i.e., human, object, and interaction proposal) ranging from $K_g$ to $K_g\times3$ (**4 to 12**) for the geometric group and from $K_s$ to $K_s\times3$ (**2 to 6**) for the semantic group. Notably, over 96% of samples in V-COCO and over 94% in HICO-DET contain no more than 12 entities, so geometric group of 4 to 12 can cover the majority of scenarios. Meanwhile, the size of semantic group used in the paper also aligns with [ref3]'s finding, which underscores that the majority of the interactive dialogue involving 10 or more participants is produced by only the top 4-5 contributors. Hence, we deem it unnecessary to train an estimator for neighbor count assessment, as this could introduce instability or optimization challenges. The above discussion will be incorporated into Sec. 3.5 and 4.3.
>
> [ref3] Group discussion as interactive dialogue or as serial monologue: The influence of group size. Psychological science, 2000.
>
> **Q5: How is the proximity score trained and supervised?**
>
> **A5**: Our apologies for the lack of clarity. The linear layer, used to compute the proximity score by fusing the centroid distance and IoU between two bounding boxes, is not explicitly supervised but is trained in an end-to-end manner based on the final HOI loss. To clarify, the manuscript will be revised in Sec. 3.3 as follows:
>
> > A simple linear layer is then employed to compute the proximity score $s_{i,j}$ based on the central distance $dis_{i,j}$ and the Intersection-over-Union $IoU_{i,j}$ between bounding boxes, without requiring additional supervision. This results in a proximity score matrix, where lower scores indicate closer spatial proximity. For each entity, we select the $K_g$ neighbors  with the lowest scores to construct its geometric neighbor set $N^g_i$.

---

> > ### Comment · Reviewer_H8t4 · 2025-08-07
> > **After Reading the Response**
> >
> > I appreciate your responses, which have helped clarify my initial questions. I'll increase the rating.
> >
> > Regarding the benchmarking point, I’d like to clarify that my original suggestion was not to request additional experiments on new datasets, but rather to encourage a more targeted evaluation that better highlights the key improvements of your proposed method (though, I do appreciate the additional dataset and results). This is similar in spirit to how the NVI dataset was used in [Ref2] to illustrate the strengths of the proposed approach.

---

> > > ### Author Response · Authors · 2025-08-07
> > > **Thanks for the response**
> > >
> > > Thank you for your thoughtful review and for taking the time to consider our rebuttal. We are delighted to hear that the additional information addressed your concerns, prompting a rise in the rating. Your feedback is invaluable in improving the quality of our paper and we are committed to incorporating your suggestions for a new version that reflects the changes. As part of our continued efforts, we will prioritize the development of a more targeted evaluation benchmark that better captures the strengths of our proposed method.

---

### Official Review · Reviewer_oLM7 · 2025-07-01

**Clarity:** 3
**Significance:** 2
**Originality:** 2
**Rating:** 4
**Confidence:** 3

**Summary:**

This paper introduces a framework for Human-Object Interaction detection that re-examines relation modeling from a group perspective.

The authors propose that current HOI methods neglect the inherent collective and non-pairwise social attributes within human-centric scenarios.

To address this, GroupHOI leverages two visual attraction principles: geometric proximity and semantic similarity. Geometric proximity is exploited by grouping humans and objects into distinct clusters using a learnable proximity estimator based on spatial features derived from bounding boxes. Semantic similarity is incorporated by enhancing a transformer-based interaction decoder with semantic-aware local cues derived from HO-pair features.

The paper presents extensive experiments on the HICO-DET and V-COCO benchmarks, claiming superior performance over state-of-the-art methods.

**Questions:**

Clarification on novelty vs. existing graph-based methods. Please provide a more detailed and concrete discussion distinguishing GroupHOI's geometric and semantic grouping mechanisms from prior graph-based HOI detection methods (e.g., DRG, SCG, and other works that use spatial/semantic relationships for graph construction). What are the fundamental differences in how groups are formed and how information is propagated that makes GroupHOI a novel approach rather than an incremental improvement?

Elaborating "social attributes". Could you expand on how the proposed geometric and semantic grouping truly captures "inherent social attributes" beyond basic visual co-occurrence or similarity? Are there specific architectural components or loss functions that encourage the model to learn more abstract social cues, or is "social attributes" primarily a high-level interpretation of visual grouping?

Comprehensive SOTA comparison. Please provide an updated comparison table that includes very recent state-of-the-art methods, especially those published in the last three years, to firmly establish GroupHOI's competitive performance. If the current SOTA claim is indeed challenged by other works, please re-evaluate and clarify the contribution in light of these stronger methods.

More details. Could the authors provide more fine-grained implementation details? This could include specific architectural diagrams for the geometric-aware local transformer and semantic-aware interaction decoder, exact configurations for the learnable proximity estimator and pooling operator, and any non-standard training tricks or hyperparameter tuning procedures.

Experimental analysis on the cost and burden of the proposed method.

**Ethical Concerns:**

["NO or VERY MINOR ethics concerns only"]

**Final Justification:**

Rebuttal address my main concern, I will increase my score.

**Limitations:**

The paper lacks discussion over the additional cost as a limitation.

**Paper Formatting Concerns:**

No significant formatting issues here.

**Quality:**

3

**Strengths And Weaknesses:**

Pros:

1. Presentation. The paper is generally well-written and easy to follow. The methodology is explained logically, and the figures, particularly Figure 1 illustrating relation modeling paradigms, are helpful in understanding the core ideas.


2. Novel perspective on social attributes for HOI detection. The emphasis on learning social attributes of human interactions in groups is an interesting and valid perspective. It attempts to capture more complex, collective patterns in visual scenes, which aligns with how humans naturally categorize and understand groups.

3. Structured Grouping Principles: The definition of two visual attraction principles – proximity and similarity – to construct groups is clear and easy to follow. It makes the contribution of the paper more understandable.

Cons:

1. Originality for overlap with existing graph-based methods. The claim of novelty for the proposed solutions, specifically the geometric-aware local transformer and semantic-aware interaction decoder, needs further justification. As is shown in Section 3.2, these solutions bear significant resemblance to previous two-stage HOI detection methods that focus on graph modeling. The paper's distinction from existing graph-based methods like DRG (ECCV 2020) , SCG (ICCV 2021), SKGHOI (ICDMW 2023) utilizing spatial relationships and semantic similarities to build graphs for HOI classification, is not sufficiently highlighted or critically analyzed. The differences seem incremental rather than a significant departure from established graph-based approaches.


2. Weak link of the solution to the motivation of "social attributes in a group of people". The connection between the proposed technical solutions (geometric-aware local transformer and semantic-aware interaction decoder) and the overarching idea of "learning social attributes of human interactions in a group" appears tenuous. While geometric proximity and semantic similarity can contribute to group formation, the paper doesn't deeply elaborate on how these mechanisms specifically capture or represent "social attributes" beyond basic visual grouping. The discussion remains at a low-level feature aggregation rather than a high-level social understanding.

3. Claim of SOTA performance. The claim of sota) performance may not be entirely accurate. The paper appears to miss some recent works published in the past three years that might achieve higher performance, especially on the V-COCO dataset. For instance, VIL (ACMMM 2023) and CQL (CVPR 2023) and some more recent works obtain stronger performance than GroupHOI on V-COCO, which could challenge the paper's main empirical claim. A more comprehensive and up-to-date comparison is necessary.

4. Reproducibility for insufficient implementation details. While Section 3.5 provides some implementation details, they may not be granular enough to ensure full reproducibility of the work.

5. The additional modules may brought significant computational overhead. A experimental analysis on the cost is very important.

6. In the last paragraph of introduction, the authors seem to confuse the results for V-COCO and HICO-Det.

---

> ### Author Rebuttal · Authors · 2025-07-31
>
> We thank reviewer oLM7 for the valuable time and constructive feedback. We provide point-to-point response below.
>
> ***
>
> **Q1: The novelty needs further justification.**
>
> **A1**：We apologize for the confusion. In fact, our method is significantly different from previous two-stage graph-based HOI approaches in the following aspects:
>
> **1) Motivation**
>
> Previous two-stage HOI detection methods, such as DRG [ref1] , SCG [ref2] and SKGHOI [ref3], primarily construct graphs to perform the interaction reasoning between humans and objects. In contrast, our method constructs groups to explore the **latent social attributes** based on two visual attraction principles, enhancing feature representation of each proposal by aggregating information from its neighbors.
>
> **2) Formulation**
>
> Prior work constructs a **complete bipartite graph** between objects and humans with size $N_h \times N_o$, where $N_h$ and $N_o$ denote the number of object and human proposals, respectively. Our method constructs **localized geometric and semantic graphs** with sizes only  $(N_h + N_o) \times K_g$ and $N_i \times K_s$, respectively, with $N_h$, $N_o$, and $N_i$ denoting the number of human, object, and interaction proposals, and $K_g$, $K_s$ representing the geometric and semantic group sizes. Notably, $K_g \ll N_h/N_o$ and $K_s \ll N_i$, highlighting the efficiency of our method.
>
> **3) Implementation**
>
> Prior two-stage methods relied on Graph Neural Network (GNN) architectures, employing iterative message passing via message functions to perform interaction reasoning. Our method can be directly integrated into one-stage frameworks. It defines neighbor sets defined by geometric proximity and semantic similarity, and employs simple yet effective operators (i.e., self-attention and max-pooling) to enhance the feature representations of human/object and interaction proposals.
>
> In summary, our method fundamentally differs from prior graph-based methods in motivation, formulation, and implementation, which underscores the novelty of our work.  We will revise the Sec. 2 of the manuscript to highlight our novelty.
>
> [ref1] DRG: Dual Relation Graph for Human-Object Interaction Detection. ECCV, 2020.
>
> [ref2] Spatially Conditioned Graphs for Detecting Human-Object Interactions. ICCV, 2021.
>
> [ref3] SKGHOI: Spatial-Semantic Knowledge Graph for Human-Object Interaction Detection. ICDMW, 2023.
>
>
> **Q2: Weak link to the motivation of "social attributes in a group of people"**.
>
> **A2**：Thank you for pointing this out. We acknowledge the potential ambiguity in our use of the term "social attributes", which we intended to denote collective behaviors observed in real-world interactions (i.e., multiple humans engaging in joint activities).
> Inspired by this, we approach relation modeling from a group perspective, conceptually grounded in the Gestalt principles of Proximity and Similarity [ref4], which describe the mechanisms underlying human visual organization. To operationalize these principles, we formulate **geometric proximity groups** and **semantic similarity groups**, and propose two modules (i.e., geometric-aware local transformer and semantic-aware interaction decoder) to integrate these groups into the HOI detection pipeline. To enhance clarity, the Abstract and Sec. 1 of the manuscript will be revised as follows:
>
> Revision for Abstract:
>
> > However, current HOI methods primarily focus on pairwise relationships, overlooking the social attributes in a group of humans (i.e., multiple humans engaging in joint activities).
>
> Revision for Sec. 1:
>
> > Despite achieving strong performance, these methods are still confined to predefined pairwise relations, leaving the inherent social attributes which exhibit collective patterns (i.e.,  multiple entities engaging in joint activities) unexplored.
>
> > we construct social groups guided by the Gestalt grouping principles of Proximity and Similarity: First, the geometric proximity principle, i.e., the tendency for individuals to form groups with those physically close to, is implemented by constructing geometric groups based on the central distance and Intersection over Union (IoU) of bounding boxes. Second, the semantic similarity principle, i.e., the tendency for individuals to affiliate with those sharing visual characteristics or behavioral patterns, is operationalized through building semantic groups of interaction proposals based on similarity of their feature embeddings.
>
> [ref4] Ellis, W. D. (2013). A source book of Gestalt psychology. Routledge.
>
>
> **Q3: More recent sota works for comparison**.
>
> **A3**：Thanks for your suggestion. Per your request, we compare our model with more recent works published in the past three years.
>
> | Method              | Full (DF) | Rare (DF) | Non-Rare(DF) | $AP^{S1}_{role}$ | $AP^{S2}_{role}$ |
> | ------------------- | --------- | --------- | ------------ | ---------------- | ---------------- |
> | VIL [ACMMM23]       | 34.21     | 30.58     | 35.30        | 65.3             | 67.7             |
> | CQL [CVPR23]        | 35.36     | 32.97     | 36.07        | 66.4             | 69.2             |
> | HOIGen [ACMMM24]    | 34.84     | 34.52     | 34.94        | -                | -                |
> | CEFA [ACMMM24]      | 35.00     | 32.30     | 35.81        | 63.5             | -                |
> | **GroupHOI (Ours)** | **36.70** | **34.86** | **37.26**    | **65.0**         | **66.0**         |
>
> For V-COCO, GroupHOI exhibits competitive performance but remains inferior to STIP [ref5], VIL [ref6], and CQL [ref7]. However, on HOI-DET, our method surpasses these approaches by a solid margin, especially for rare interactions. We attribute this to the enhanced capabilities of GroupHOI in more complex interaction scenarios, as evidenced by: HOI-DET contains **600** HOI interactions (vs. **293** in V-COCO), an average of **4** interactions (vs. **3** in V-COCO) and **6** entities per sample (vs. **4** in V-COCO). Furthermore, we also provide additional validation on NVI dataset [ref8], which contains an average of **5**  social interactions involving a group of humans (from **2** to **12**). The experimental results on NVI val also support this claim:
>
> | Method              | mR@25     | mR@50     | mR@100    | AR        |
> | :------------------ | :-------- | :-------- | :-------- | :-------- |
> | m-QPIC              | 56.89     | 69.52     | 78.36     | 68.26     |
> | m-CDN               | 55.57     | 71.06     | 78.81     | 68.48     |
> | m-GEN-VLKT          | 50.59     | 70.87     | 80.08     | 67.18     |
> | NVI-DEHR [ECCV24]   | 54.85     | 73.42     | 85.33     | 71.20     |
> | **GroupHOI (Ours)** | **55.67** | **76.73** | **87.16** | **73.19** |
>
> The  results and discussion on V-COCO will be updated in Sec. 4.2. We will also add Sec. 4.4 for the analysis of results on NVI dataset [ref8]  in the revised manuscript.
>
> [ref5] Exploring structure-aware transformer over interaction proposals for human-object interaction detection. CVPR, 2022.
>
> [ref6] Improving Human-Object Interaction Detection via Virtual Image Learning. ACMMM, 2023.
>
> [ref7] Category Query Learning for Human-Object Interaction Classification. CVPR, 2023.
>
> [ref8] Nonverbal Interaction Detection. ECCV, 2024.
>
> **Q4: More implementation details.**
>
> **A4**：Thanks for your suggestion. More implementation details and pseudo code can be found in Sec. A and Sec. D of the Supplementary Material, respectively. The Sec. A has been expanded to include additional details as follows:
>
> > To address the dimensional mismatch between positional embeddings (256) and the interaction decoder (768), we expand the positional embeddings by stacking them three times. During training, we replace the Focal Loss used in HOICLIP [ref9] with Asymmetric Loss [ref10] to better handle the long-tail distribution of HOI categories.
>
> Futhermore, as stated in L17 and L245, we commit to making our code publicly available upon acceptance.
>
> [ref9] Hoiclip: Efficient knowledge transfer for hoi detection with vision-language models. CVPR, 2023.
>
> [ref10] Asymmetric loss for multi-label classification. ICCV, 2021.
>
> **Q5: Model efficiency analysis.**
>
> **A5**：Following your suggestion, we conducted a detailed comparison of GroupHOI with other HOI methods in terms of Parameters (Params), Floating Point Operations (FLOPs), and Frames Per Second (FPS) as follows:
>
> | Method              | Params↓   | FLOPs↓     | FPS↑      | Full (DF) |
> | ------------------- | --------- | ---------- | --------- | --------- |
> | PPDM [CVPR20]       | 194.9M    | 121.63G    | 15.28     | 21.10     |
> | GEN-VLKT [CVPR22]   | 41.9M     | 60.04G     | 26.18     | 33.75     |
> | HOICLIP [CVPR23]    | 66.1M     | 104.68G    | 19.57     | 34.59     |
> | ViPLO [CVPR23]      | 118.2M    | 41.35G     | 14.89     | 34.95     |
> | **GroupHOI (Ours)** | **79.2M** | **83.67G** | **16.42** | **36.70** |
>
> where Full (DF) represents the results on Default evaluation setting of  HICO-DET. Compared to HOICLIP [ref11], which shares the same network architecture as GroupHOI except for the GEO and SEM modules, GroupHOI introduces a slight increase in parameter count and a marginal reduction in FPS, yet achieves a substantial performance improvement, surpassing HOICLIP by +2.11 mAP on the Full setting of HICO-DET. Moreover, when compared with ViPLO [ref12], a graph-based method, under the same conditions, GroupHOI delivers superior accuracy while using fewer parameters and achieving faster inference, highlighting the efficiency of our model design (see Q1 **Implementation**). We will supplement Sec. 4.3 with this analysis in the revised manuscript.​
>
> [ref11] Hoiclip: Efficient knowledge transfer for hoi detection with vision-language models. CVPR, 2023.
>
> [ref12] Viplo: Vision transformer based pose-conditioned self-loop graph for human-object interaction detection. CVPR, 2023.
>
> **Q6: The results for V-COCO and HICO-Det are confused**.
>
> **A6**：We apologize for this mistake. It has been corrected in the revision.

---

> > ### Comment · Reviewer_oLM7 · 2025-08-05
> >
> > The rebuttal address my main concern, I will increase my score.

---

> > > ### Author Response · Authors · 2025-08-05
> > > **Thanks for the response**
> > >
> > > Thank you for your thoughtful review and for taking the time to consider our rebuttal. We are delighted to hear that the additional information addressed your concerns, prompting a rise in the rating. Your feedback is invaluable in improving the quality of our paper and we are committed to incorporating your suggestions for a new version that reflects the changes.

---

### Decision · Program_Chairs · 2025-09-17

**Decision:**

Accept (poster)

**Comment:**

This paper proposes GroupHOI, a novel framework for Human-Object Interaction (HOI) detection that revisits relation modeling from a group perspective. Instead of focusing on pairwise relationships, it propagates contextual information based on two principles: geometric proximity and semantic similarity.

The key strength of this work is its innovative and intuitive approach to modeling HOI through group dynamics, which is technically sound and well-motivated. The proposed geometric and semantic grouping mechanisms are effectively integrated into a strong baseline, achieving state-of-the-art results on the complex HICO-DET and V-COCO benchmarks.

Initial weaknesses included questions about novelty compared to prior graph-based methods, the need for more SOTA comparisons, and justification for certain design choices like fixed group sizes. During the rebuttal, the authors provided thorough responses, including additional experiments, efficiency analysis, and clarifications that successfully addressed most of the reviewers' concerns. Although the reported contributions appear somewhat overstated as concerned by reviewer 8R1p, other reviewers raised their scores, acknowledging the authors' detailed feedback.

After careful consideration, the AC recommends accepting this paper.